# Boundary layer models for calving marine outlet glaciers

Christian Schoof[1], Andrew D. Davis[2], and Tiberiu V. Popa[1]

[1]Department of Earth, Ocean and Atmospheric Sciences, University of British Columbia, 2020-2207 Main Mall, Vancouver, BC, V6T 1Z4, Canada
[2]Department of Aeronautical Engineering, Massachusetts Institute of Technology, Cambridge, MA, USA

*Correspondence to:* Christian Schoof (cschoof@eoas.ubc.ca)

**Abstract.** We consider the flow of marine-terminating outlet glaciers that are laterally confined in a channel of prescribed width. In that case, the drag exerted by the channel side walls on a floating ice shelf can reduce extensional stress at the grounding line. If ice flux through the grounding line increases with both, ice thickness and extensional stress, then a longer shelf can reduce ice flux by decreasing extensional stress. Consequently, calving has an effect on flux through the grounding line by regulating the length of the shelf. In the absence of a shelf, it plays a similar role by controlling the above-flotation height of the calving cliff. Using two calving laws, one due to Nick *et al* based on a model for crevasse propagation due to hydrofracture, and the other simply asserting that calving occurs where the glacier ice becomes afloat, we pose and analyse a flowline model for a marine-terminating glacier by two methods: direct numerical solution and matched asymptotic expansions. The latter leads to a boundary layer formulation that predicts flux through the grounding line as a function of depth to bedrock, channel width, basal drag coefficient, and a calving parameter. By contrast with unbuttressed marine ice sheets, we find that flux can decrease with increasing depth to bedrock at the grounding line, reversing the usual stability criterion for steady grounding line location. Stable steady states can then have grounding lines located on retrograde slopes. We show how this anomalous behaviour relates to the strength of lateral versus basal drag on the grounded portion of the glacier, and to the specifics of the calving law used.

## 1 Introduction

In the theory of laterally unconfined marine ice sheet flow, a standard result is that flux through the grounding line is an increasing function of bedrock depth (Weertman, 1974; Thomas and Bentley, 1978; Schoof, 2007a). This leads to the conclusion that grounding lines can have stable steady states only when the ice sheet bed has sufficiently steep down-flow slope (Fowler, 2011; Schoof, 2012): a slight advance in grounding line position into deeper water leads to an increase in flux through the grounding line, causing the ice sheet retreat back to its original position. Analogously, a slight retreat leads to a reduction in flux through the grounding line and a re-advance of the ice sheet.

There are a number of mechanisms that can alter the flux-to-bedrock-depth relationship. These include the appearance of 'hoop stresses' in an ice shelf fringing the ice sheet (see Pegler and Worster, 2012; Pegler, 2016, though these may require unrealistically large ice shelves), the fact that bedrock elevation can actually change due to loading and unloading of the lithosphere (Gomez et al., 2010), thermomechanically mediated changes in basal friction (Robel et al., 2014, 2016), and lateral drag due to geometrical confinement of the flow into a channel (Dupont and Alley, 2005; Jamieson et al., 2012). The latter is

probably the most significant mechanism; when ice flows through a channel, drag can be generated by the side walls of the channel. Drag at the channel side walls of the floating ice shelf reduces the extensional stress acting at the grounding line and, therefore, reduces the grounding line flux.

Goldberg et al. (2009) and Gudmundsson et al. (2012) demonstrate that sidewall drag can lead to the formation of stable steady states with grounding lines on upward-sloping (or 'retrograde') beds. Both papers have channels of uniform width and fix the edge of the ice shelf, which suggests the following physics: for a steady-state grounding line on an upward-sloping bed, a slight retreat in grounding line position will cause an increase in ice thickness at the grounding line. In isolation, this would lead to increased discharge and continued ice sheet retreat. However, the retreat in grounding line position also leads to a longer shelf, which therefore experiences more lateral drag and reduces extensional stress at the grounding line. This process is known as 'buttressing' and, by itself, would lead to reduced discharge and a re-advance of the ice sheet. Which of the two mechanisms dominates presumably depends on the specifics of the channel and the fixed shelf front position.

An open question is whether an evolving calving front can lead to a similar stabilization, as we are no longer guaranteed that a retreat in the grounding line position leads to the same increase in shelf length and, therefore, to the same increase in lateral drag. To answer that question conclusively, we would need a universal 'calving law' that can robustly predict the location of the calving front. Such a calving law is currently not available.

We investigate how two particular calving laws that are relatively widely used in models for tidewater glaciers affect buttressing in a simplified flowline model. The ice flow model itself lacks the sophistication of models that resolve the cross-channel dimension. Instead, it relies on a parameterization of lateral drag in terms of the mean along-channel velocity (Dupont and Alley, 2005; Nick et al., 2010; Jamieson et al., 2012; Hindmarsh, 2012; Pegler et al., 2013; Robel et al., 2014, 2016; Pegler, 2016). The chief advantages of the model are that it allows flux through the grounding line to be computed rapidly as a function of ice thickness through the use of a boundary layer theory (Schoof, 2007a) and that the role of different physical mechanisms becomes comparatively easy to trace. Future work will be required to address whether our results are reproduced qualitatively by more sophisticated (and more computationally intensive) models, and we hope that this paper can motivate such work.

The rationale for the calving models used here is described in greater detail in section 2.2. One calving law simply states that calving occurs at the local flotation thickness at the grounding line. The calving front is at flotation when calving occurs but no floating ice shelf ever forms. We use this calving law as a simple reference case that should give results analogous to previous work on unconfined marine ice sheet flow (Schoof, 2007a), where the formation of a floating shelf has no effect on flux through the grounding line. The other calving law that we investigate is the 'CD' calving law due to Nick et al. (2010). Simulations of outlet glaciers in Greenland with this calving law have predicted stabilization of grounding lines on areas of upward-sloping bed (Nick et al., 2013), suggesting that it may indeed predict a relationship between flux and bedrock depth that differs from theories for unconfined marine ice sheet flow. We do, however, stress that our aim is not an exhaustive survey of all calving models. We anticipate that the analysis presented below can be applied to other calving models, but doing so is beyond the scope of our paper.

## 2 Model

### 2.1 Ice flow

We consider a flowline model for a rapidly sliding, channelized outlet glacier with a parameterized representation of lateral drag. The model has the same essential ingredients as those in Dupont and Alley (2005), Jamieson et al. (2012), Nick et al. (2010), Hindmarsh (2012), Pegler et al. (2013) and Pegler (2016). Fig. 1 shows the physical domain. Mass accumulates over the glacier and is advected seaward by ice flow. Mass is ultimately lost by flow across the grounding line and eventual calving of icebergs. Our notation is summarized in a table given in the supplementary material: let $x$ be the along-flow coordinate and $t$ time, while $u(x,t)$ and $h(x,t)$ are width-averaged ice velocity and thickness, respectively. If $b(x)$ is bed elevation and $w(x)$ the width of the outlet channel, each assumed constant in time, then we model force balance and mass conservation as

$$2(\bar{B}h|u_x|^{1/n-1}u_x)_x - \bar{B}'w^{-1/n-1}h|u|^{1/n-1}u - \theta C|u|^{m-1}u - \rho_i g\left(1 - (1-\theta)\rho_i/\rho_w\right)h(h_x + \theta b_x) = 0 \tag{1a}$$

$$wh_t + (wuh)_x = w\left(a - (1-\theta)m\right) \tag{1b}$$

for $0 < x < x_c(t)$, where subscripts $x$ and $t$ denote partial derivatives. Here, $\rho_i$ and $\rho_w$ are the densities of ice and water, respectively, and $g$ is acceleration due to gravity, while $a$ is surface mass balance and $m$ is the melt rate at the base of the floating ice. The indicator function $\theta$ is given by

$$\theta(x) = \begin{cases} 1 & \text{if } \rho_i h(x) \geq -\rho_w b(x), \\ 0 & \text{otherwise;} \end{cases} \tag{1c}$$

in other words, $\theta = 1$ if and only if the ice thickness is above flotation and the glacier is grounded.

Note that we have included the melt rate $m$ in (1) for completeness only. While the numerical code included with the supplementary material permits computations of steady state solutions in section 3.2 and of the boundary layer problem described in section 4.2 with a non-zero basal melt rate under floating ice, a full exploration of the extended parameter space would make this paper unmanageable. In what follows, we set $m = 0$ throughout, and will address the effect of incorporating non-zero $m$ in a separate paper.

The parameters $\bar{B}$ and $n$ are the usual parameters in the Glen's law rheology for ice (Paterson, 1994). We neglect any complications associated with the dependence of ice viscosity on temperature or moisture content, and treat $\bar{B}$ as well as $n$ as constant. $C$ is a drag coefficient in a power-law basal friction law, with $m$ the corresponding exponent (e.g. Budd et al., 1979; Fowler, 1981); on theoretical grounds, it is often assumed that $m = 1/n$. Note that other friction laws have also been considered in boundary layer models for unconfined ice sheets (Tsai et al., 2015). The basal drag term only applies where ice is grounded, corresponding to $\theta = 1$. For simplicity, we neglect the possibility that $C$ may depend on additional degrees of freedom such as basal water pressure or temperature.

The second term $\bar{B}'w^{-1/n-1}h|u|^{1/n-1}u$ is a parameterization of lateral drag, with $\bar{B}'$ another constant. We assume that lateral drag is exerted on both, grounded and floating ice. A more sophisticated treatment of lateral drag would require a

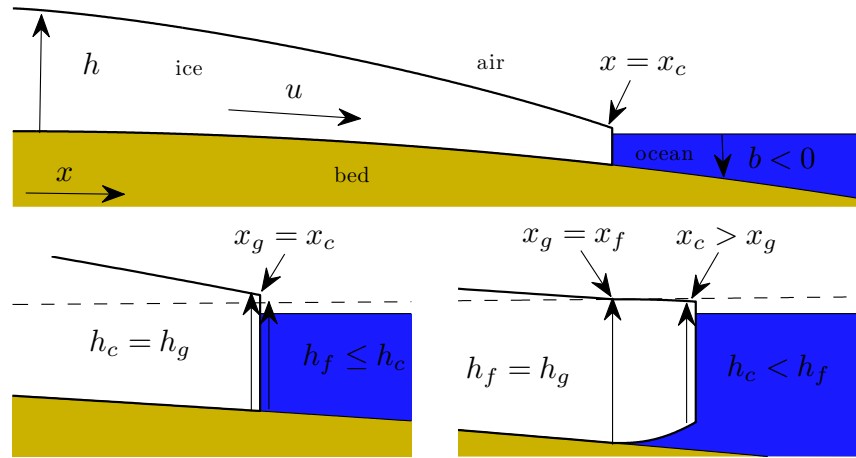

**Figure 1.** Schematic of the model domain and variables used. Not shown is the lateral dimension: the glacier occupies a channel of width $w$, potentially variable along the glacier flowline. The upper ice surface shown is at an elevation $s = (1 - (1 - \theta)\rho_i/\rho_w)h + \theta b$, the lower surface at $l = s - h$. The lower two diagrams are enlargements of the vicinity of a grounded terminus (left) and a floating terminus (right), illustrating the meaning of calving front position $x_c$, grounding line position $x_g$, calving front height $h_c$, grounding line thickness $h_g$ and flotation thickness $h_f$. The dashed line indicates where the upper ice surface would need to be at a given position $x$ in order for the ice to be just about to float.

domain with two horizontal dimensions, and an additional equation representing force balance in the direction perpendicular to the channel axis (e.g. Wearing et al., 2015).

It is worth noting however that (1a) becomes accurate in one of two mutually incompatible limits: (i) a wide channel where lateral drag vanishes (this is the one-dimensional flow case previously studied in e.g. Schoof (2007a)), or (ii) a long and narrow

glacier where extensional stress is insignificant and there is no significant flow transverse to the channel axis. By 'extensional stress', we mean the non-cryostatic part of normal stress on a vertical surface placed across the flow, that is, $2\bar{B}|u_x|^{1/n-1}u_x$. In the narrow-channel case, assuming no slip at the channel side walls, (1) predicts the correct width-averaged velocity if we put (see also e.g. Raymond, 1996, for details on flows dominated by lateral shear)

$$\bar{B}' = (n+2)^{1/n}2^{(n+1)/n}\bar{B};\tag{1d}$$

smaller values of $\bar{B}'$ can be justified if there is actual sliding at the lateral margins of the ice. We use (1a) even when neither of the two limits above apply. As discussed in Pegler (2016), this is a simplification that works reasonably well and allows at least semi-analytical progress to be made. The simplicity of the model has also led to a large number of authors adopting versions of it. We proceed in that spirit, analysing the model at face value.

We denote the glacier terminus by $x_c(t)$; this is the location where ice cover ends. Since $x_c$ is a free boundary, two boundary

conditions are required. One is needed to close the elliptic problem (1a) and another to determine the evolution of $x_c$. The

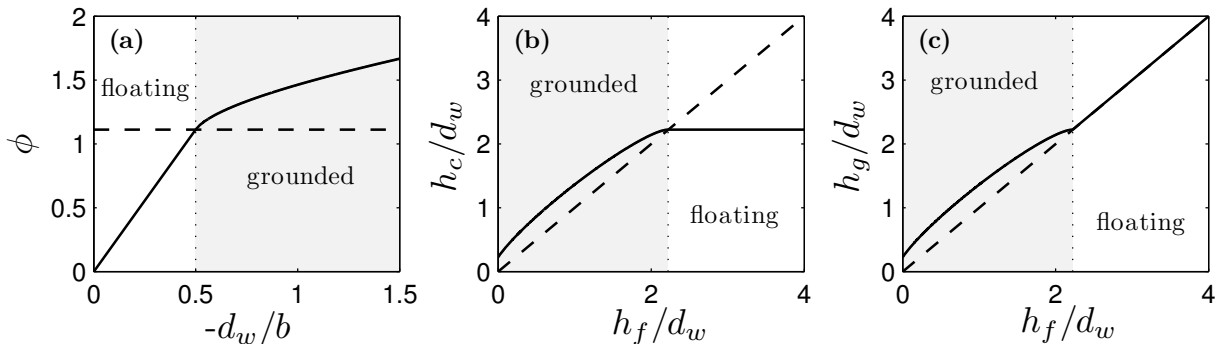

**Figure 2.** Calving laws. In all three panels, a solid line refers to the CD calving law, a dashed line to the FL law. The grey shaded region refers to parts of parameter space where the calving front is grounded for the CD law, white background to a floating calving front. $r = 0.9$ throughout. Panel (a): the calving-front-thickness-to-bedrock-depth ratio $\phi$ as a function of normalized water depth $-d_w/b$. For the FL law, the ratio is simply $r^{-1}$. Panel (b): normalized calving front thickness $h_c/d_w$ as a function of normalized flotation thickness $h_f/d_w = -b/(rd_w)$ at the grounding line. Panel (c): normalized grounding line thickness $h_g/d_w$ as a function of normalized flotation thickness $h_f/d_w$.

former is a condition on extensional stress at the ice front (e.g. Schoof, 2007b, appendix B):

$$2\bar{B}h|u_x|^{1/n-1}u_x = \rho_i\left(1 - (1-\theta)\rho_i/\rho_w\right)h^2/2 - \theta\rho_w gb^2/2 \qquad \text{at } x = x_c(t). \tag{1e}$$

We take the second condition to be a 'calving law'. While a stress condition is sufficient to close the force balance model (1a), a calving model can be understood as fixing the free boundary location. The next section describes the different choices of
calving laws used here.

## 2.2   Calving model

The process of calving remains relatively poorly understood, but several calving laws have been developed on theoretical grounds. Our aim is to illustrate how different calving laws can lead to qualitatively, rather than quantitatively, different dynamics in the outlet glacier. We consider two possible calving laws. The first is the 'CD' model due to Nick et al. (2010), and
the second is a calving law that states that ice breaks off when the glacier reaches its flotation thickness. To streamline our notation, we refer to the latter as the 'FL' calving law.

The CD model works based on the assumption that water in surface crevasses affects the depth to which those crevasses can penetrate. When they penetrate deeply enough to connect with basal crevasses, calving occurs. When they do not, there is no calving and the ice front simply moves at the velocity of the ice. Algebraic manipulation of the Nick et al. (2010) CD model
shows that connections with basal crevasses occur instantly in the model when ice thickness is at (or below) a value $h_c$. In

other words, the evolution of the calving front $x_c$ satisfies

$$\text{either } h = h_c \qquad\qquad\qquad\qquad\qquad\qquad \text{at } x = x_c \text{ if } \dot{x}_c \leq u(x_c), \tag{1f}$$

$$\text{or } \dot{x}_c = u \qquad\qquad\qquad\qquad\qquad\qquad \text{at } x = x_c \text{ if } h > h_c, \tag{1g}$$

where the dot indicates differentiation with respect to time. Note that the domain lies to the left of $x_c$, so $\dot{x}_c < u$ implies that ice is removed by calving. $h_c$ itself can be written as a function of the crevasse water depth $d_w$ and of local bedrock depth $-b$. This function can be expressed as

$$h_c = -b\phi(-d_w/b) \tag{1h}$$

where
$$\phi(-d_w/b) = \begin{cases} 2(\rho_w/\rho_i)(-d_w/b) & -d_w/b < 1/2 \\ \nu + \sqrt{\nu^2 - (\rho_w/\rho_i)} & -d_w/b \geq 1/2, \end{cases} \tag{1i}$$

and $\nu$ is

$$\nu = 1 + [(\rho_w/\rho_i) - 1](-d_w/b).$$

The function $\phi$ is the ratio of calving front thickness to depth to bedrock; the form of $\phi$ is illustrated in panel a of Fig. 2.

As an alternative to the CD model, in which the function $\phi$ is defined through (1i), we consider the FL law, in which the glacier calves 'at flotation.' This means that calving occurs when $h = -(\rho_w/\rho_i)b$ at the calving front, but not when $h$ is larger. This is easy to incorporate into the calving framework (1f)–(1h) above: we simply have to replace the definition of $\phi$ in (1i) with the simpler

$$\phi = \rho_w/\rho_i. \tag{1j}$$

This condition is effectively what applies in previous work on marine ice sheet flow without sidewall drag as considered in e.g. Schoof (2007a).

Note that Nick et al. (2010) do not formulate their calving law directly in the form (1f)–(1i); a derivation of the thickness condition (1f) based on their formulation is given in the supplementary material. There are two cases in (1i): $-d_w/b < 1/2$ corresponds to a floating terminus and $-d_w/b \geq 1/2$ to a grounded terminus. Note that $\phi$ and $h_c$ are continuous (in fact, continuously differentiable, as shown in Fig. 2) at $-d_w/b = 1/2$, where calving occurs at the critical floatation thickness $h_c = (\rho_w/\rho_i)b$. Once the calving front is afloat, $h_c$ no longer depends on bedrock depth: when $-d_w/b < 1/2$, we simply have $h_c = 2(\rho_w/\rho_i)d_w$. In other words, for a fixed water depth parameter $d_w$ and sufficiently large bedrock depths the Nick et al. (2010) CD model is actually a calving law that simply states that ice breaks off a floating glacier shelf at a critical thickness that is determined purely by the parameter $d_w$ (panel b of Fig. 2).

To complete the notation for our model, we also define the grounding line position $x = x_g$. For a glacier with a floating shelf, this is the point $x = x_g(t)$ at which $\theta$ changes discontinuously:

$$h(x_g(t), t) = -(\rho_i/\rho_w)b(x_g(t)); \tag{1k}$$

For the FL calving law, there is no floating shelf; the grounding line and calving front coincide. The CD calving model goes further, permitting cases where there is no floating ice shelf and the flotation condition $h = -(\rho_i/\rho_w)b$ is attained nowhere along the glacier, including at the calving front. To keep our terminology as simple as possible, we identify the grounding line in that case with the terminus location,

$$x_g = x_c. \tag{1l}$$

For later convenience, we also define the ice thickness $h_g$ at the grounding line and the flotation thickness $h_f$ at the grounding line through

$$h_g = h(x_g(t),t), \qquad h_f = -(\rho_i/\rho_w)b(x_g(t)). \tag{1m}$$

For glaciers with a floating shelf, we always have $h_g = h_f$, and for that reason, existing theories for marine ice sheets generally do not make a distinction between $h_g$ and $h_f$ (e.g. Schoof, 2007a). The distinction becomes relevant when there is no floating ice shelf, in which case we only have $h_g \geq h_f$. We will use $h_f$ frequently below, as it is a simple function of bedrock depth at the grounding line, and therefore determined purely by the geometry of the glacier channel. $h_g$ additionally depends on the calving process as shown in panel c of Fig. 2.

In our view, the CD model is a cartoon version of the linear elastic fracture mechanics explored in by Weertman (1973, 1980) and van der Veen (1998a, b). These papers consider the 'mode 1' (Zehnder, 2012) propagation of vertical cracks into ice under tensile (extensional) stresses. This is done by computing stress intensity factors at the crack tips from known Green's functions for parallel-sided elastic slabs with cracks penetrating from the upper or lower surfaces, accounting for the pressure exerted by water in the cracks, and applying a fracture toughness criterion. The CD model by contrast assumes that extensional stress increases with depth in the ice in a linear, cryostatic fashion. The model then computes crevasse penetration as being the distance from the upper and lower surfaces at which that extensional stress becomes sufficiently negative (that is, sufficiently compressive) to overcome the pressure exerted by water at the same depth. The CD model therefore does not compute stress with the same level of sophistication as the papers by Weertman (1973, 1980) and van der Veen (1998a, b), but follows the same basic approach of computing crevasse propagation based on a known ice geometry, extensional stress and crevasse water pressure, and it has the advantage of tractability.

The basic method in van der Veen (1998a, b) in principle allows for a constraint to be computed that links ice thickness, applied extensional stress, crevasse water level and fracture toughness at the moment that surface and basal crevasses together first penetrate through the entire ice thickness. Given that extensional stress is a function of ice thickness through (1e), this constraint could be converted into a criterion for the thickness $h_c$ at which calving occurs, giving a more sophisticated version of the Nick et al. (2010) CD model. However, the papers by van der Veen do not deal with the case in which both, surface and basal crevasses are present and interact with each other (so the relevant Green's functions are not given), and he does not explicitly compute a condition for calving that could be put in the form (1f). As a result, we confine ourselves to the simpler CD model here.

Note that there is one inconsistency in the calving law at small $-b/d_w$: here the CD law predicts values of $h_c < d_w$ (see Fig. 2(b)). Obviously, this implies a greater water level in surface crevasses than the ice thickness, which is physically impossible.

We are led to conclude that, for small enough $-b/d_w$ (which will later correspond to small enough values of the parameter $\Lambda$ defined in section 4.2), the calving model continues to lead to computable results but breaks down physically. Note that having a flotation height (equal to $-\rho_w b/\rho_i$) that is smaller than $d_w$ is not problematic: the calving front thickness can be greater than the flotation height, and therefore allow for such large water depths.

5  A second practical pitfall of the CD model is that it predicts no calving at all if $d_w = 0$ and surface crevasses are free of water. It is possible that this is an artifact of the simple representation of stress in the CD model, where the tensile stress driving crevasse propagation is assumed to have the same dependence on depth below the ice surface regardless of whether a crevasse is present or not. In reality, the formation of crevasses that penetrate through a significant fraction of the ice shelf leads to extensional stress becoming more concentrated around the crack tips than for shallow crevasses (see for instance Fig. 4 of

10 van der Veen (1998a)). This represents a positive feedback on crack propagation, and could lead to calving even for the case of water-free surface crevasses (see also Weertman, 1980). In addition, the stress fields considered by Weertman (1973, 1980), van der Veen (1998a, b) and Nick et al. (2010) are relatively simple and apply only at distances significantly greater than a single ice thickness from the calving front. In the calving of shorter, taller icebergs, torques near the calving front (Hanson and Hooke, 2000; Ma et al., 2017) may allow calving when purely extensional stresses experienced further upstream do not.

15  More recently, others have extended the linear elastic fracture mechanics approach of Weertman (1973, 1980) and van der Veen (1998a, b) to include effects such as the role distributed damage due to the formation of microcracks in initiating crevasse formation, the blunting of cracks tips due to viscous deformation, and the presence of significant torques near the calving front (Krug et al., 2014; Mobasher et al., 2016; Jiménez et al., 2017; Yu et al., 2017). The complexity of these processes however makes them difficult to parameterize in a model that does not resolve the scale of individual crevasses, and we do not consider

20 them here.

  The Nick et al. (2010) CD calving model, along with the work of Weertman (1973, 1980) and van der Veen (1998a, b), is based on tensile failure. We can contrast this with the shear failure model of Bassis and Walker (2011) (see also Bassis and Jacobs (2013) and Ma et al. (2017)). The CD model requires $d_w > 0$ and predicts calving for any $h$ below the value given by (1f), instantaneously removing all parts of the glacier shelf that are too thin. By contrast, the shear failure model of Bassis and

25 Walker (2011) predicts that calving will start at a critical calving front thickness and not occur below that thickness, so the inequality in (1g) would need to be reversed. It also predicts that once initiated, the calving front will continue to fracture as it moves into thicker ice inland. This is the basis of the catastrophic calving cliff instability mechanism for marine ice sheet collapse advocated by Pollard et al. (2015), but cannot be captured by an analogue of (1f). It is clear that ice sheets whose calving cliff is larger than the critical thickness for shear failure simply cannot persist: they are guaranteed to disintegrate

30 completely or to stabilize in some shape where the calving front thickness is below the critical thickness for shear failure, and the shear failure model by itself does not provide a timescale for that disintegration. We exclude such shear failure from consideration here and focus purely on the CD calving model.

  Even taking the Nick et al. (2010) CD model at face value, as we do here, the sensitivity to the parameter $d_w$ remains problematic. In fact, one of our results below will be that flux through the grounding line is more sensitive to $d_w$ than to any

35 other model parameter. At present, we do not have a surface hydrology model that can predict $d_w$. It is plausible that a future

hydrology model could compute a water table height near the calving cliff ($d_s - d_w$ if measured relative to the local ice surface, where $d_s$ is the depth of surface crevasses as discussed in the supplementary material) rather than using $d_w$ itself. Such a model would likely be based on drainage being driven by gradients in hydraulic head, but this awaits future development. We persist with the basic Nick et al. (2010) model, treating $d_w$ as given.

## 3  Solution of the model

### 3.1  Non-dimensionalisation

In the remainder of this paper, we will consider the problem (1) in dimensionless form. The purpose of doing so is two-fold. Non-dimensionalisation (i) reduces the number of free parameters and (ii) allows systematic approximations based on the relatively small size of some dimensionless parameters. We assume that we know scales $[a]$ for accumulation rate and $[x]$ and $[w]$ for glacier length and width, respectively. We choose scales $[u]$, $[h]$, and $[t]$ based on the balances

$$\bar{B}'[w]^{-1/n}[h][u]^{1/n} = \rho_i g[h]^2/[x], \qquad [u][h] = [a][x], \qquad [u][t] = [x].$$

We define dimensionless variables as $u = [u]u^*$, $h = [h]h^*$, $x = [x]x^*$, $t = [t]t^*$, $x_c = [x]x_c^*$, and also put

$$\varepsilon = \frac{\bar{B}[w]^{1/n+1}}{2\bar{B}'[x]^{1/n+1}}, \qquad \gamma = \frac{C[w]^{1/n+1}[u]^{m-1/n}}{B'[h]}, \qquad \lambda = \frac{d_w}{[h]}, \qquad r = \frac{\rho_i}{\rho_w}, \tag{2}$$

$$a^* = \frac{a}{[a]}, \qquad w^* = \frac{w}{[w]}, \qquad b^* = \frac{b}{[h]}. \tag{3}$$

Dropping asterisks on the dimensionless variables immediately, we obtain

$$4\varepsilon(h|u_x|^{1/n-1}u_x)_x - w^{-1/n-1}h|u|^{1/n-1}u - \gamma\theta|u|^{m-1}u - (1 - r + \theta r)h(h_x + \theta b_x) = 0, \tag{4a}$$

$$wh_t + (wuh)_x = wa \tag{4b}$$

for $0 < x < x_c(t)$, with $\theta$ the indicator function for flotation

$$\theta = 1 \quad \text{if } rh \geq -b, \qquad \theta = 0 \quad \text{otherwise} \tag{4c}$$

and the boundary conditions at the terminus being

$$4\varepsilon h|u_x|^{1/n-1}u_x = (1-r)h^2/2 + \theta\left(r^2h^2 - b^2\right)/(2r) \qquad\qquad \text{at } x = x_c(t), \tag{4d}$$

and either

$$h = -b\phi(-\lambda b^{-1}) \qquad\qquad \text{at } x = x_c \text{ if } \dot{x}_c \leq u(x_c), \tag{4e}$$

$$\text{or } \dot{x}_c = u \qquad\qquad \text{at } x = x_c \text{ if } h > -b\phi\left(-d_w/b\right), \tag{4f}$$

with $\phi$ given by (1i) for the CD calving model, or by (1j) (which states that $\phi \equiv r^{-1}$) for the FL calving law.

## 3.2 Direct numerical solution

The system (4) can be solved numerically as posed. In this paper, we focus on solutions of the steady-state version of the problem by a shooting method, which provides a straightforward alternative to a solution by more established time-stepping methods. As our method has not been used previously in this context, we sketch it here for completeness; results are presented at the end of this section and in Figs. 3 and 4.

We can write the steady state problem as a four-dimensional, first-order autonomous system of differential equations if, in addition to $h$, we define the phase space variables $q$, $\sigma$ and $\chi$ through

$$q = uhw, \qquad \sigma = |u_x|^{1/n-1}u_x, \qquad \chi = x. \tag{5}$$

For technical reasons associated with singular behaviour of the steady state problem near ice divides, we also define a new independent variable $\eta$ through

$$x_\eta = q$$

to obtain a first order system of differential equations from (4):

$$h_\eta = -h^2 w(\chi)|\sigma|^{n-1}\sigma + hw(\chi)a(\chi) - hqw'(\chi)w(\chi)^{-1} \tag{6a}$$

$$\begin{aligned}
\sigma_\eta ={}& (4\epsilon)^{-1}h^{-1/n}|q|^{1/n+1}w(\chi)^{-2/n-2} + \gamma\theta h^{-m-1}|q|^{m+1}w(\chi)^{-m} \\
&+ (4\epsilon)^{-1}\left[1 - (1-\theta)r\right]\left[-h^2 w(\chi)|\sigma|^{n-1}\sigma + hw(\chi)a(\chi) - hqw'(\chi)w(\chi)^{-1}\right] \\
&- (4\epsilon)^{-1}\theta qb'(\chi) + h|\sigma|^{n+1}w(\chi) - \sigma w(\chi)a(\chi) + \sigma w'(\chi)w(\chi)^{-1}
\end{aligned} \tag{6b}$$

$$q_\eta = qaw(\chi) \tag{6c}$$

$$\chi_\eta = q \tag{6d}$$

with $\theta(h,\chi) = 1$ if $h > -b(\chi)/r$ and $\theta = 0$ otherwise; here $a$, $w$ and $b$ are treated as prescribed functions, and the prime simply indicates their first derivative.

We assume there is an ice divide at $x = 0$, where $u = q = 0$. Technically, the ice divide then becomes a fixed point of the system (6) approached as $\eta \to -\infty$, at which $(h, \sigma, q, \chi) = (h_0, [a(0)/h_0]^{1/n}, 0, 0)$ with the ice divide thickness $h_0 > 0$ not known *a priori*. The trick is to determine the value of $h_0$ for which the boundary conditions at the glacier terminus are satisfied by means of a shooting method. Given $h_0$, the fixed point has a unique orbit that emerges from it. In other words, if $h_0$ is known, then the solution to (6) can be computed uniquely. A constraint on $h_0$ therefore arises from imposing the boundary conditions at the glacier terminus, which are of the form

$$4\varepsilon h\sigma = (1-r)h^2/2 + \theta\left(r^2 h^2 - b(\chi)^2\right)/(2r), \qquad h = -b\phi(-\lambda b(\chi)^{-1}) \tag{6e}$$

at some finite $\eta = \eta_c$. (6e) is dealt with simply by integrating along the orbit until the first condition is satisfied. The second then acts as a single constraint on the degree of freedom $h_0$ that uniquely determines the solution. The code used to compute solutions here is included in the supplementary material.

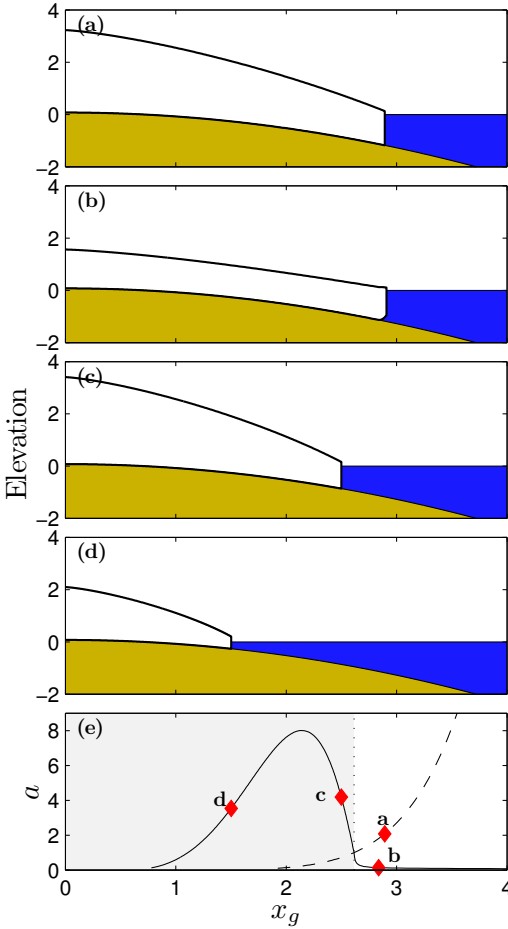

**Figure 3.** Steady state profiles, with $w \equiv 1$, $n = 1/m = 3$, $r = 0.9$, $\gamma = 0$, $\varepsilon = 0.01$, $\lambda = 7.0304 r \varepsilon^{n^2/(n+1)^2}$, $b(x) = \varepsilon^{n^2/(n+1)}(1 - 2x^2)$. The apparently contrived form in which $\lambda$ and $b$ are written is designed to make comparison with section 4.2 simpler, where the corresponding variables $\Lambda$ and $B$ are then easy to extract. Panel (e) shows grounding line positions $x_g$ against surface accumulation rate $a$ for the CD calving model (solid line, grounded where the background is shaded grey) and FL calving law (dashed line). Panels (a)–(d): steady state profiles, same colour scheme as in Fig. 1, same horizontal axis as in (e). Panel (a): The FL calving law, $a = 2.08$. Panels (b)–(d): CD calving law with $a = 0.134$ (b), 4.18 (c), 3.54 (d). Red diamonds in panel (e) refer to the steady state in the panel indicated by the letter label.

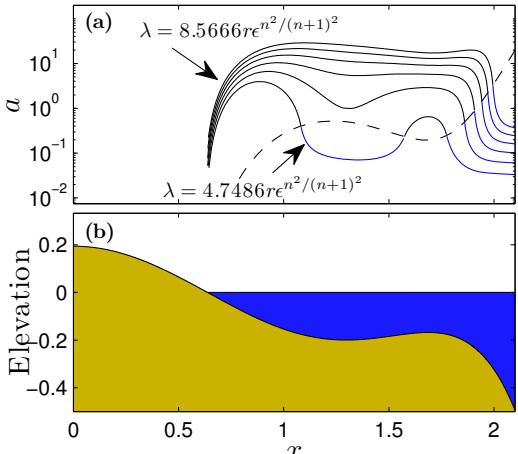

**Figure 4.** Steady state profiles, with $w \equiv 1$, $n = 1/m = 3$, $r = 0.9$, $\gamma = 0$, $\varepsilon = 0.001$, , $b(x) = \varepsilon^{n^2/(n+1)}(9.4665 - 28.3710x^2 + 13.3975x^4 - 1.97021x^6)$). Panel (b) shows the geometry of the bed, which is a scaled version of the sixth-order polynomial bed shape used in Schoof (2007b). Panel (a) shows grounding line positions $x_g$ against surface accumulation rate $a$ (note the logarithmic scale) for the FL model (dashed line) and the CD calving model (solid line, shown in black where the calving front is grounded, blue where it is afloat) at values of $\lambda = 4.7486r\varepsilon^{n^2/(n+1)^2}$, $5.5122r\varepsilon^{n^2/(n+1)^2}$, $6.2758r\varepsilon^{n^2/(n+1)^2}$, $7.0304r\varepsilon^{n^2/(n+1)^2}$, $7.8030r\varepsilon^{n^2/(n+1)^2}$, $8.5666r\varepsilon^{n^2/(n+1)^2}$. To identify the curves, note that for a given grounding line position, larger values of $\lambda$ invariably correspond to larger values of $a$.

Fig. 3 shows a synthetic example, not based on trying to emulate any specific glacier geometry. Parameter values are given in the figure caption: here, we hold constant the bed and channel geometries, material properties such as $m$, $n$ and $r$ as well as the calving parameter $\lambda$ fixed and vary accumulation rate $a$. Importantly, the bed slopes downward monotonically in $x$. For 'unbuttressed' glaciers subject to basal but not lateral drag (e.g. Schoof, 2007b), this would lead to steady state grounding line
position $x_g$ increasing monotonically with accumulation rate $a$: increased accumulation inland must be balanced by increased discharge of ice across the grounding line, which happens for unbuttressed glaciers when the grounding line moves into deeper water. The most notable feature in panel b is that this behaviour persists if we use the FL model (dashed line). For the Nick et al. (2010) CD model, we see a partial reversal of this behaviour: for accumulation rates larger than a certain value, there appear to be no steady state solutions at all. For smaller accumulation rates, there are two steady state solutions: (i) a large ice
sheet for which $x_g$ shrinks as $a$ increases, and (ii) a small ice sheet for which $x_g$ increases with $a$. The larger solution branch also contradicts existing understanding of marine ice sheet dynamics, precisely because an increase in surface mass balance causes the grounding line to retreat into shallower water. Such steady states are likely to be unstable (see Schoof, 2012, and section 6 below)

Fig. 4 shows analogous calculations to Fig. 3, but for an overdeepened bed shape based on that used in Schoof (2007b).
For the FL model, we invariably see that an increase in accumulation rate makes the grounding line advance on a downward slope, and retreat on a retrograde slope. This is again analogous to the unbuttressed case studied in Schoof (2007b), where the grounding line is then unstable when located on an upward slope. For the CD model, the behaviour becomes more complicated.

We see that the grounding line can either advance or retreat with increasing accumulation rate, on both the downward- and upward-sloping parts of the bed. Qualitatively, shallow water depths at the grounding line are more commonly associated with the standard, 'unbuttressed' behaviour (that is, an increase in accumulation tends to cause the grounding line to advance on downward slopes, and retreat on retrograde slopes). The reverse behaviour is typically associated with larger water depths at the grounding line. We also see that a decrease in $\lambda$ leads to the 'reverse' behaviour being observed down to shallower water depths at the grounding line, and in particular, through more of the overdeepened section. Note also that the solid (CD model) solution curves in Fig. 3 end at finite values of $a$ at $x_g = 0.6386$, the location where water depth goes to zero: as shown in panel (b) of Fig. 2, an oddity of the CD model is that it predicts a non-zero calving front thickness even when the water depth is zero, and hence there is a non-zero calving flux even where the 'grounding line' is on dry land; we have only computed solutions where the grounding line remains in the water.

Our aim in what follows is to explain the results in Figs. 3–4 using the same boundary layer approach as in Schoof (2007a). In particular, we will show that flux through the grounding line can be computed to leading order in the parameter $\varepsilon$ as a function depth to bedrock and channel width at the grounding line, as well as of the calving parameter $\lambda$, friction coefficient $\gamma$ and the remaining physical parameters $(r, m, n)$. Given such a relationship, it is then possible to determine how the grounding line location in a steady state depends on accumulation rates, purely by balancing net accumulation over the domain with outflow of ice through the grounding line.

## 4   Approximation: a small lateral aspect ratio

### 4.1   A local force balance version of the model

If we take $\varepsilon$ as defined in (2) with $\bar{B}'$ given by (1d), we have

$$\varepsilon = (n+2)^{1/n} 2^{1/n} \left( \frac{[w]}{[x]} \right)^{(n+1)/n}.$$

In other words, $\varepsilon$ is a measure of the lateral aspect ratio $[w]/[x]$. A narrow channel ensures that $\varepsilon$ is small, which is the basis for our approximation scheme. With $\varepsilon$ small, we can neglect gradients of the depth-integrated extensional stress in (4a) (that is, gradients of $4h|u_x|^{1/n-1}u_x$) everywhere except close to the terminus, and find

$$-w^{-1/n-1}h|u|^{1/n-1}u - \gamma|u|^{m-1}u - h(h_x + b_x) = 0, \tag{7a}$$

$$wh_t + (wuh)_x = wa. \tag{7b}$$

For the case of $m = 1/n$ (which arises naturally from theories of hard-bed sliding, see e.g. Weertman (1957), Fowler (1981)), we get a diffusive model for ice thickness evolution,

$$wh_t - \left[ \frac{w^{n+2}h^{n+1}}{(h + \gamma w^{(n+1)/n})^n} |(h+b)_x|^{n-1}(h+b)_x \right]_x = wa. \tag{8}$$

This is essentially analogous to 'shallow ice' models in ice sheet flow (Fowler and Larson, 1978): we have a local balance of forces, and an ice flux that depends on ice thickness and ice surface slope (see also Kowal et al., 2013). Other choices of $m > 0$

also imply an ice flux $uh$ that is an increasing function of width $w$, thickness $h$ and surface slope $-(h_x + b_x)$, but that flux cannot be computed in closed form.

## 4.2 The grounding line boundary layer

The model (7) holds everywhere except near the grounding line and in the floating ice shelf. Following Schoof (2007a), we can use the method of matched asymptotic expansions (Holmes, 1995) to capture the behaviour of ice flow in that region. This requires us to rescale the dimensionless model (4) to bring back extensional stress at leading order while maintaining an $O(1)$ ice flux $q = uh$. The appropriate rescaling turns out to be

$$X = \varepsilon^{-n/(n+1)}(x - x_g), \qquad T = t, \qquad H = \varepsilon^{-n^2/(n+1)^2}h, \qquad U = \varepsilon^{n^2/(n+1)^2}u.$$

By contrast with Schoof (2007a), we also have to include a potentially non-zero ice shelf length here, so we put

$$X_c = \varepsilon^{-n/(n+1)}(x_c - x_g).$$

We treat $H$ and $U$ as functions of $(X,T)$ and $X_c$ as a function of $T$. The rescaling in $H$ implies that ice thickness at the grounding line must be small compared with the interior of the glacier. If there is a floating portion, the glacier must however also reach its flotation thickness at the grounding line. We assume that the glacier is at least near flotation if it has a calving front that remains above flotation. This implies that bed elevation must be small compared with ice thickness in the interior. Specifically, we rescale

$$B = \varepsilon^{-n^2/(n+1)^2}b$$

and assume that $B \sim O(1)$; the analogous case of laterally unconfined flow discussed in Schoof (2007a) also requires shallow bed topography. In addition, we assume that thickness $b$ and width $w$ change significantly only over length scales associated with the glacier length as a whole. Over the short length scale associated with the boundary layer coordinate $X$, we treat $B = \varepsilon^{-n^2/(n+1)^2}b(x_g)$ and width $W = w(x_g)$ as constant. These additional constraints are again analogous to those made in Schoof (2007a), and imply that we should treat $b$ and $b_x$ as small in the outer problem (7). With $B$ constant at leading order in the boundary layer, we can also define a scaled flotation thickness

$$H_f = -r^{-1}B,$$

which we will use throughout the rest of the paper as a proxy for water depth to bedrock.

In order for the rescaling in $H$ above to be consistent, we also require that the calving front thickness be similarly small. This turns out to require that $\lambda \sim O(\varepsilon^{n^2/(n+1)^2})$, and we define the calving parameter

$$\Lambda = r^{-1}\varepsilon^{-n^2/(n+1)^2}\lambda$$

5   assuming that $\Lambda = O(1)$; all this implies is that water depths in surface crevasses are not so large as to create calving cliff heights much larger than the expected depth to bedrock at the grounding line.

The result is a boundary layer model at leading order in $\varepsilon$. We do not give the detailed derivation here but merely state its form:

$$4(H|U_X|^{1/n-1}U_X)_X - W^{-1/n-1}H|U|^{1/n-1}U - \gamma\varepsilon^{n[1-n(m+1)]/(n+1)^2}\theta|U|^{m-1}U - [1-(1-\theta)r]\,HH_X = 0, \tag{9a}$$

$$(HU)_X = 0, \tag{9b}$$

for $X < X_c$ where

$$\theta = 1 \quad \text{for } H \geq H_f, \qquad \theta = 0 \quad \text{otherwise.} \tag{9c}$$

The additional boundary condition at the calving front takes the form

$$4H|U_X|^{1/n-1}U_X = (1-(1-\theta)r)H^2/2 - \theta r H_f^2/2 \qquad\qquad \text{at } X = X_c, \tag{9d}$$

$$H = rH_f\phi\left(\Lambda H_f^{-1}\right) \qquad\qquad \text{at } X = X_c \tag{9e}$$

Note that the boundary layer is in a pseudo-steady state. This is again analogous to Schoof (2007a); the time scale for dynamic adjustment of ice thickness in the boundary layer and of calving front position relative to the grounding line is much shorter than the time scale relevant to the evolution problem (7) (see also Pattyn et al., 2012). We emphasize that there is no assumption here that the glacier as a whole is in steady state.

In order to make the balances in (9) work with $\varepsilon \ll 1$, we have to deal with the remaining coefficient that contains a power of $\varepsilon$ in (9). We now make further assumptions about the physics of the flow near the grounding line. Our fundamental assumption will be that lateral drag $-w^{-1/n-1}h|u|^{1/n-1}u$ plays a leading order role in force balance at the grounding line, but that the floating ice shelf, if it exists, is not so long as to fully buttress the grounding line. By this, we mean that the depth-integrated extensional stress $4h|u_x|^{1/n-1}u_x$ is comparable in magnitude to $h^2$ all the way up to the grounding line, as is the case at the terminus $x_c$ by dint of the boundary condition (4d), but that sidewall drag cannot be neglected.

In that physical regime (termed a 'distinguished limit', in which all physical processes are potentially active), we have to assume that the basal drag coefficient in (9a) and the calving coefficient in (9e) are both of $O(1)$, meaning that the parameter $\Gamma$ defined through

$$\Gamma = \varepsilon^{n[1-n(m+1)]/(n+1)^2}\gamma \tag{9f}$$

is of $O(1)$. We confine our analysis to parameter regimes where this is the case. Note that with $m > 0$ and $n \geq 1$, this implies strictly speaking that $\gamma \ll 1$, and basal friction upstream of the boundary layer is formally small in the parameter regime we are considering.

Asymptotic matching is the mathematical formalism by which the boundary layer problem and the 'outer' problem (7) for the dynamics of the bulk of the glacier are connected (Holmes, 1995). With the assumptions on $\Gamma$ in place, this leads to so-called matching conditions between the boundary layer and the outer problem:

$$\lim_{X\to-\infty} UH = Q = \lim_{x\to x_g^-}(-w^{n+1}h|h_x|^{n-1}h_x), \qquad W^{-1/n-1}Q|U|^{1/n-1} \sim -(Q/U)(Q/U)_X,$$

$$U \to 0 \qquad \text{as} \qquad X \to -\infty. \tag{9g}$$

Here $Q$ is the flux at the boundary of the domain of the outer problem, to be determined through the solution of the boundary layer problem. Physically, the first condition states that the flux near the grounding line in the 'outer' problem is the flux that enters the boundary layer at its upstream end. The second condition states that near that upstream end of the boundary layer, extensional stress gradients have become insignificant and flux is given by a shallow ice type formula (with $U = Q/H$, the condition can be re-written as $Q \sim -W^{n+1}H|H_X|^{n-1}H_X$, the appropriate local-force-balance formula in our case). Lastly, the third condition states that velocities in the interior of the boundary layer are large compared with those in the rest of the glacier. Structurally, the boundary layer problem above is very similar to that in Schoof (2007a), with additional physics due to lateral shearing and calving accommodated at the cost of a more complicated formulation.

From the perspective of the model (8) for the dynamics of the glacier as a whole, the purpose of the boundary layer is to provide the relevant boundary conditions at $x = x_g$. As (8) is a diffusion model for $h$, it requires two boundary conditions at any moving boundary. Where a floating portion exists, a condition on $h$ at the grounding line arises straightforwardly from (1k); for a grounded calving front, an equivalent condition is provided by (4e).

As in previous work (Schoof, 2007a), we can show that the second boundary condition takes the form of a flux condition that can be found by solving the boundary layer problem: the problem (9) has a solution only if $Q$ satisfies a functional relationship with flotation thickness $H_f$, width $W$, friction coefficient $\Gamma$ and the calving parameter $\Lambda$. It is important to emphasize again that $H_f$ need not be the ice thickness at what we have termed the 'grounding line'. Instead, $H_f$ is the flotation thickness there, determined purely by bedrock depth, and ice thickness equals $H_f$ at the grounding line only if the glacier has a floating shelf or is at the point of forming one. The flotation thickness $H_f = -B/r$ is of course prescribed for any given grounding line position, as is the channel width $W$. We therefore end up with ice flux as a function of grounding line position, basal drag coefficient $\Gamma$ and the calving parameter $\Lambda$, itself a proxy for water depth in surface crevasses.

We give additional detail on how to compute that relationship between flux, geometry and model parameters in appendix B and in the supplementary material, and the code used to solve the problem is also included in the supplementary material. Importantly, we are able to show the relationship takes the form

$$Q = W H_f^{n+1} G_\Lambda \left( \frac{H_f}{\Lambda}, \frac{\Gamma W^{(nm+n+m+1)/(n+1)}}{\Lambda^{2-nm}}, n, m, r \right). \tag{10a}$$

The practical use of this form is that it reduces the complexity of the flux formula: for a given set of constants $n$, $m$ and $r$, what we primarily need to calculate is the dependence of flux on the first two arguments of the function $G_\Lambda$. For the FL law, we can in fact go further and use the fact that flux cannot depend on the now redundant parameter $\Lambda$ to simplify the expression to

$$Q = W H_f^{n+1} G_\Gamma \left( \frac{\Gamma W^{(nm+n+m+1)/(n+1)}}{H_f^{2-nm}}, n, m, r \right). \tag{10b}$$

## 5   Solutions of the boundary layer problem

Equation (10a) allows us to collapse solutions for flux onto a one-parameter family of plots for each of the two calving laws considered (the FL and CD calving laws). Specifically, we can plot $Q/(W\Lambda^{n+1})$ against $H_f/\Lambda$ for fixed values of $\Gamma W^{(nm+n+m+1)/(n+1)}/\Lambda^{2-nm}$. Roughly speaking, we can think of this as plotting flux $Q$ against flotation thickness $H_f$ for

different values of the basal drag coefficient. Solutions are plotted in this way in panel a of Fig. 5. The black curves signify solutions with vanishing basal friction ($\Gamma = 0$), while coloured curves show solutions with non-zero values of $\Gamma$ as specified in the figure caption. The dashed line in each case corresponds to the FL model, while the solid line corresponds to the CD calving model.

As already suggested by the steady-state solutions to the full problem in Fig. 3, Fig. 5 confirms that flux is not a mono-

tonically increasing function of flotation thickness $H_f$ for the CD model. We have what we term an anomalous flux-flotation-thickness relationship for large enough values of $H_f$: flux $Q$ actually decreases with increasing flotation thickness $H_f$ for all but relatively small $H_f$, at least for moderate or small basal drag coefficients. For large values of the basal drag coefficient (the $\Gamma W^{(n+1)/n}/\Lambda = 25$ and $125$ cases shown), the relationship between $Q$ and $H_f$ is even more complicated. We have the same anomalous flux-flotation-thickness relationship as for small basal drag while the calving front is grounded, but for a floating

calving front, we find that flux $Q$ increases again with $H_f$ (this is even clearer in panel a of Fig. 7, which is a zoomed-in version of Fig. 5). In all cases, the flux for the CD calving model approaches the same limit for large $H_f$, independently of the calving law.

By contrast, the flux always increases with flotation thickness in the calving at floatation model, just as it does in laterally unconfined marine ice sheet flow (Schoof, 2007a). In fact, equation (10b) already told us as much for the case of vanishing

basal friction coefficient $\Gamma$. Note that the flux curve for the CD model and for the FL model always have a point of intersection at $H_f/\Lambda = 2$. From the definition of $\phi$ in (1i), it is easy to see that this is the point at which the calving front is just at flotation in the CD model. Therefore, the model produces the same result as the FL model. For smaller values of $H_f/\Lambda$, the CD model has a grounded calving front, while the calving front becomes the terminus of a floating ice shelf at larger values of $H_f/\Lambda$. Note that flux in the CD model is always a decreasing function of $H_f/\Lambda$ for $H_f/\Lambda$ slightly less than the critical value of 2 for

changeover from a grounded to a floating terminus. This observation will be key to our interpretation of the physics involved in the anomalous flux-thickness relationship.

Other features of our solutions are also shown in panels a–c of Fig. 5. Each panel isolates one parameter ($H_f$, $\Lambda$, $W$) on the horizontal axis, but normalizes it as dictated by (10), and plots it against an also normalized flux (again as dictated by (10)) on the vertical axis. Apart from the dependence of $Q$ on $H_f$, panel a also shows that flux always decreases with increasing

friction coefficient $\Gamma$, while panel c shows that flux increases with channel width $W$. This holds regardless of the calving model used, and is what one expects intuitively: wider channels and lower basal drag ought to speed up ice flow and lead to larger ice discharge. Panel b shows that for the CD calving law, flux also increases with the calving parameter $\Lambda$: recall that the calving parameter $\Lambda$ is a dimensionless version of water depth in surface crevasses, and larger values of $\Lambda$ lead to taller calving cliffs

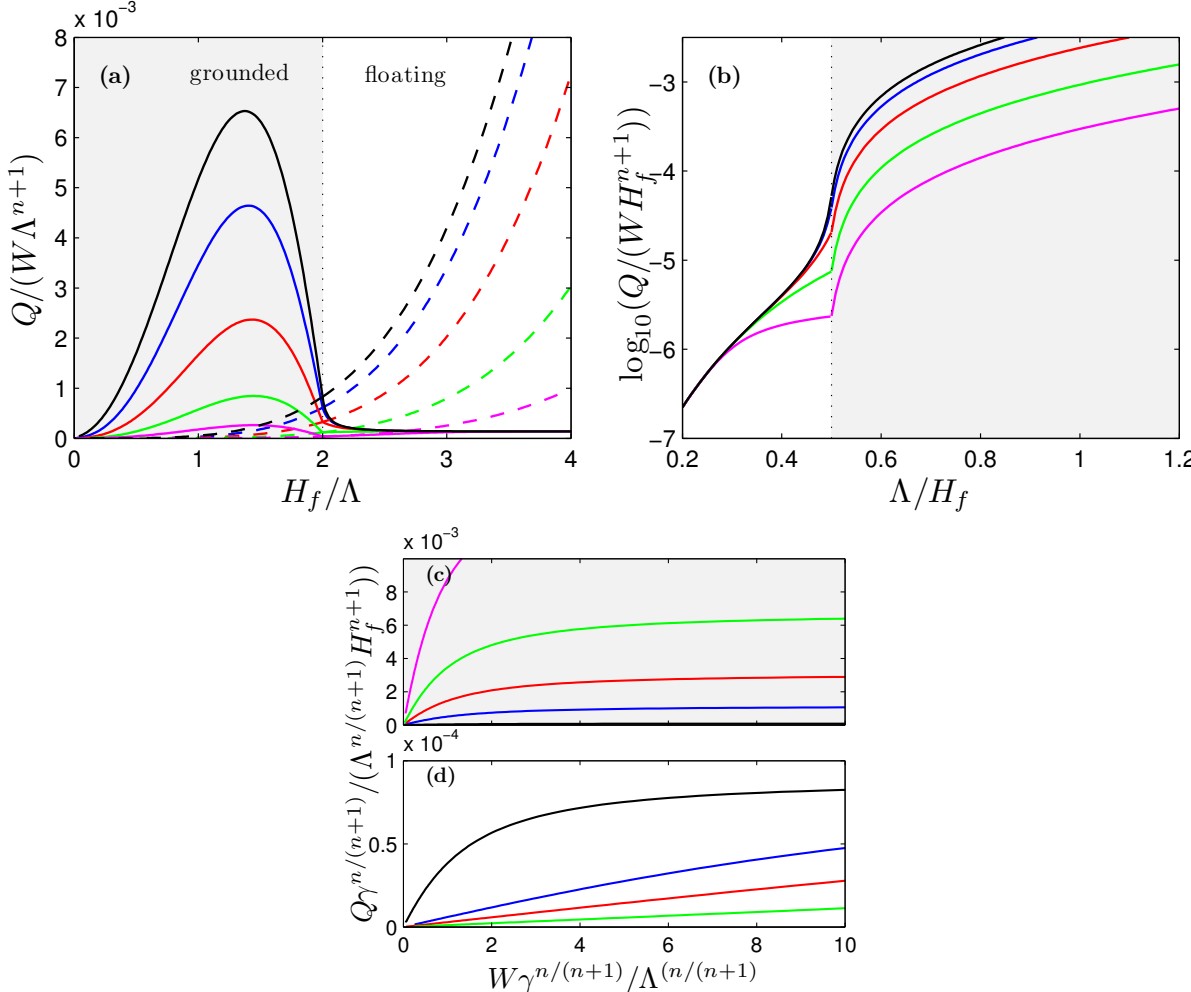

**Figure 5.** Solutions of the boundary layer problem. $r = 0.9$, $n = 1/m = 3$. Dashed lines represent the FL law, solid lines the CD calving law. Grey shading indicates a grounded calving front in the CD model, white background a floating shelf. Panel (a): normalized flux against normalized thickness at different basal friction parameter values, $\Gamma W^{(n+1)/n}/\Lambda = 0$ (black line), 1 (blue), 5 (red), 25 (green), 125 (magenta). Panel (b): logarithm of normalized flux against normalized calving parameter at different basal friction parameter values, $\Gamma W^{(n+1)/n}/H_f = 0$ (black line), 1 (blue), 5 (red), 25 (green) and 125 (magenta). Panel (c): normalized flux against normalized channel width at different (grounded) ice thickness: $H_f/\Lambda = 2$ (black), 5/3 (blue), 4/3 (red), 1 (green) and 2/3 (magenta). Panel (d): same as panel, but floating ice thickness values: $H_f/\Lambda = 2$ (black), 7/3 (blue), 8/3 (red) and 10/3 (green).

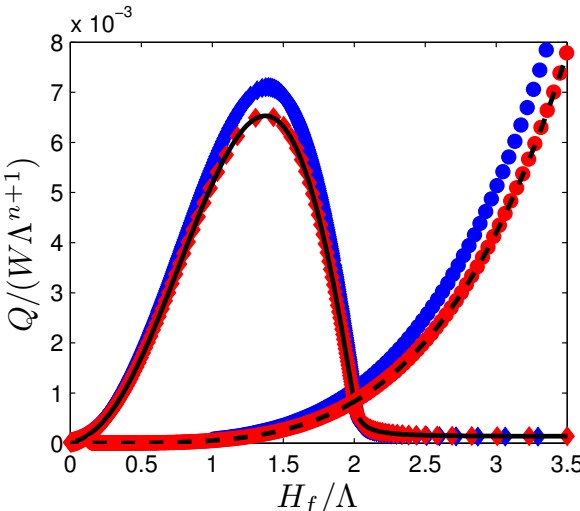

**Figure 6.** Comparison of the solution of the boundary layer problem with $\Gamma = 0$ (black lines, solid for the CD model, dashed for FL) and solutions of the steady state problem (6), solved with the parameter values given in the caption of Fig. 3. Plotted is scaled, normalized ice flux against scaled normalized ice thickness for $\varepsilon = 10^{-2}$ (blue symbols) and $\varepsilon = 10^{-4}$ (red symbols). Diamonds show solutions for the CD model, circles for the calving model.

and hence to larger extensional stresses near the grounding line. In fact, the flux is far more sensitive to changes in $\Lambda$ than in any other parameter: notice that panel b plots the logarithm of the flux on the vertical axis.

We can also confirm our boundary layer results by direct comparison with numerical solutions of the full ice flow problem, computed by the method in section 3.2. This is shown for the case of vanishing basal friction in Fig. 6. Here we use the same
5  parameter values as in Fig. 3, but for two different values of $\varepsilon$. Different solutions to the steady state problem are again obtained by varying $a$. For each $a$, we plot ice flux across the grounding line in the steady state solution, scaled as in section 4.2, against flotation thickness at the grounding line, also scaled as in section 4.2. As expected, the flux solutions obtained from the full steady state problem (6) for the CD and FL models converge to those obtained from the boundary layer problem as $\varepsilon$ is made smaller: for $\varepsilon = 10^{-4}$, the flux curves are virtually indistinguishable, confirming the accuracy of the boundary layer solution.
10   There are two aspects of the CD model flux solution that we still need to explain in more detail: (i) why flux decreases with increasing flotation thickness $H_f$ in some circumstances and (ii) why flux approaches a constant limit for large $H_f$ and so becomes independent of depth to bedrock in the channel, depending instead only on the calving parameter $\Lambda$. We turn to these problems next.

### 5.1 The role of extensional stress at the grounding line

Key to the flux-flotation-thickness relationship is that flux depends on the extensional stress at the grounding line, and that extensional stress in turn depends on the geometry of the calving front and floating ice shelf. For relatively small extensional

stresses $\Sigma$ defined by

$$\Sigma = 4|U_X|^{1/n-1}U_X$$

('small' meaning, $\Sigma$ much smaller than $H$), it is possible to derive an approximate formula for flux in terms of ice thickness $H_g = H(0)$ and extensional stress $\Sigma_g = \Sigma(0)$ at the grounding line $X = 0$. For the remainder of this section, we will use the commonly assumed friction exponent $m = 1/n$. We obtain from (9b) that

$$-H_X = HU_X/U$$

and if we neglect gradients of $H\Sigma$ in force balance, then (9a) leads to

$$|U|^{(n+1)/n} \approx H^2 U_X/(\Gamma + W^{-(n+1)/n}H).$$

Therefore, with the condition $4|U_X|^{1/n-1}U_X = \Sigma_g$, $H = H_g$ at $X = 0$,

$$Q \approx \left(\frac{1}{4}\right)^{n^2/(n+1)} \frac{H_g^{(3n+1)/(n+1)}\Sigma_g^{n^2/(n+1)}}{(\Gamma + W^{-(n+1)/n}H_g)^{n/(n+1)}}. \tag{11}$$

This formula is essentially a modification of formula (29) in Schoof (2007b), and its derivation is a translation of appendix A of Schoof (2007a) to our modified boundary layer problem. The omission of the extensional stress gradient can also be formalized
on the basis that the density difference $(1 - r)$ between ice and water is small, leading to gradients of $H\Sigma$ being negligible in the balance of forces (see the supplementary material).

For the FL model, it is easy to extract an analytical formula for flux as a function of channel width and depth to bedrock from (11). Specifically, we have $H_g = H_f$ and $\Sigma_g = (1 - r)H_f/2$, so we get

$$Q \approx \left(\frac{1-r}{8}\right)^{n^2/(n+1)} \frac{H_f^{(n^2+3n+1)/(n+1)}}{\left(\Gamma + W^{-(n+1)/n}H_f\right)^{n/(n+1)}}, \tag{12}$$

which simply generalizes formula (3.51) in Schoof (2007a) with $m = 1/n$ to the case of lateral as well as basal drag. With $\Gamma = 0$, we can also immediately recognize a version of formula (10b) with

$$Q \approx \left(\frac{1-r}{8}\right)^{n^2/(n+1)} WH_f^{n+1}. \tag{13}$$

Panels b and c of Fig. 7 shows that (12) performs well for $\Gamma = 0$ and $\Gamma = 125$. Clearly, (12) predict flux increasing with flotation thickness; this is the result of both grounding line thickness $H_g$ and extensional stress $\Sigma_g$ increasing with flotation
thickness $H_f$ at the grounding line. Next, we will use (11) to explain the anomalous behaviour with the CD calving law.

## 5.2  Grounded calving fronts

Here, we are interested in the anomalous relationship between $Q$ and the flotation thickness $H_f$. Recall that $H_f = -B/r$ is given by depth to bedrock $B$, and is therefore prescribed for a given grounding line location. The actual ice thickness $H_g$ at the grounding line is equal to $H_f$ when the glacier has a floating ice shelf or calves at flotation; for a grounded calving cliff, $H_f$

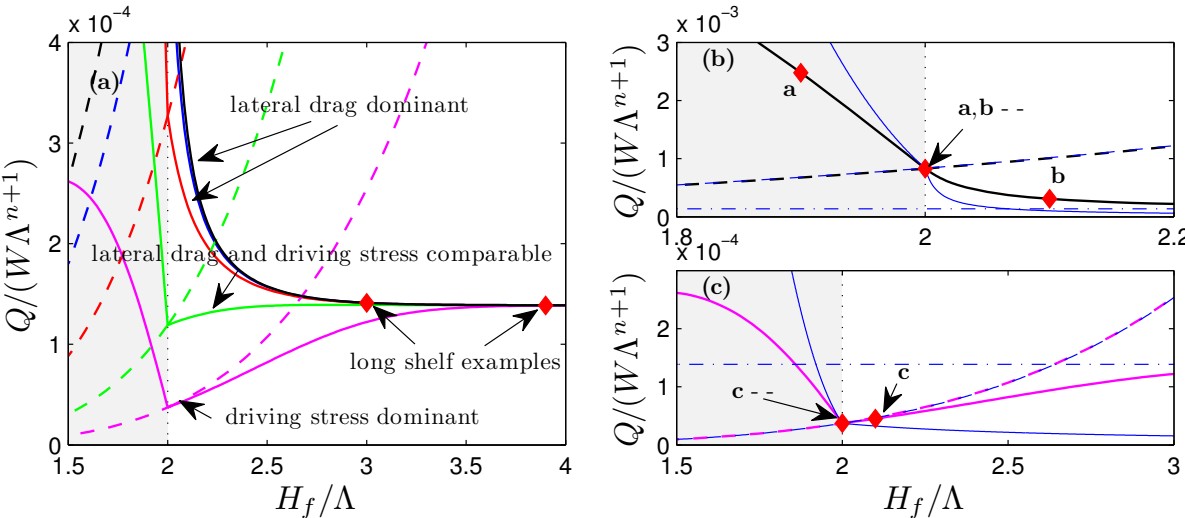

**Figure 7.** Limiting forms of ice flux near the critical ice thickness $H_f/\Lambda = 2$ and for large $H_f/\Lambda$. Panel (a): a zoomed-in version of panel (a) of Fig. 5, same colour scheme. The black, red and blue curves exhibit the anomalous relationship between flux and flotation thickness for a floating calving front, the green and magenta curves (with larger basal friction coefficients) do not. The 'dominant stress' labels refer to terms in the force balance of the shelf that balance the depth-integrated extensional stress gradient $(H\Sigma)_X$ for different basal drag coefficients $\Gamma$, see section 5.3. Red markers correspond to profiles shown in Fig. 9. Panel (b): Solution to the boundary layer problem for $\Gamma W^{(n+1)/n}/\Lambda = 0$ plotted in black (same as in panel (a)). Formula (12) is shown as a blue dashed line, formulae (16) and (20) as blue solid lines, while the dot-dashed blue line indicates the long shelf limit (24). Red markers correspond to the profiles shown in panels a and b of Fig. 8. Panel(c): same as panel (b), but with the boundary layer solution for $\Gamma W^{(n+1)/n}/\Lambda = 125$ shown in magenta, and red markers corresponding to profiles in panel (c) of Fig. 8.

may exceed $H_g$. However, as Fig. 2 shows, $H_g$ always increases with $H_f$. Equation (11) further shows that flux $Q$ increases with ice thickness $H_g$ and extensional stress $\Sigma_g$ at the grounding line. The anomalous relationship must therefore hinge on $\Sigma_g$ *decreasing* sufficiently rapidly as flotation thickness $H_f$ increases.

Note that the anomalous decrease in flux with increasing flotation thickness is most pronounced around the critical value 5 $H_f/\Lambda = 2$, where the calving front goes from grounded to floating. We can understand the behaviour of ice flux near that value by considering the effect of small perturbations in $H_f$ away from that critical value. Again, recall the actual thickness at the calving front is given by $H_c = H(X_c) = rH_f\phi(\Lambda H_f^{-1})$. Let the critical value of $H_f$ be $H_{f0} = 2\Lambda$, for which $H_c = H_{f0}$. Now consider perturbing $H_f$ slightly, say to $H_{f0} + H_f'$. We can use a first-order Taylor expansion of $\phi$ to compute the perturbed

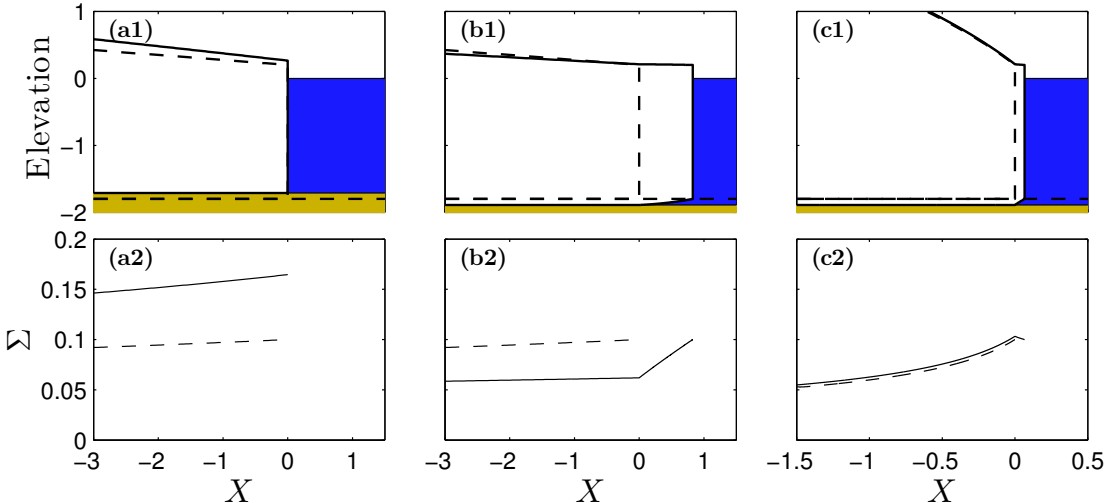

**Figure 8.** Boundary layer solutions $r = 0.9$, $n = 1/m = 3$, $\Lambda = W = 1$. Top row: boundary layer ice geometry, same colour scheme as in Fig. 1. Bottom row: the corresponding extensional stress profiles. Dashed lines correspond to the FL model with $H_f = 2$ (the 'unperturbed' flotation thickness $H_{f0}$ in sections 5.2 and 5.3), solid profiles to perturbed flotation thicknesses. Panels (a) and (b): $\Gamma = 0$ (the dashed profiles are identical), with the solid line showing (a) $H_f = 1.9$ (grounded cliff) and (b) $H_f = 2.1$ (floating calving front). Panel (c): $\Gamma = 125$, the solid line showing $H_f = 2.1$.

calving front thickness as

$$
H_c = r(H_{f0} + H'_f)\phi\left(\frac{\Lambda}{H_{f0} + H'_f}\right)
$$

$$
= r(H_{f0} + H'_f)\left[\phi\left(\frac{\Lambda}{H_{f0}}\right) - \phi'\left(\frac{\Lambda}{H_{f0}}\right)\frac{\Lambda H'_f}{H_{f0}^2} + O(H'^2_f)\right]
$$

where the prime on $\phi$ denotes an ordinary derivative. For the CD model, (1i) shows that $\phi(\Lambda/H_{f0}) = \phi(1/2) = r^{-1}$, and

5   $\phi'(\Lambda/H_{f0}) = 2r^{-1}$, so

$$
H_c = H_{f0} + O(H'^2_f) \tag{14}
$$

In other words, at linear order, a small perturbation in bedrock depth has no effect on ice thickness at the calving front in the CD model. Note that because $\phi$ in the CD model is continuously differentiable, this holds regardless of whether $H'_f$ is positive or negative, that is, regardless of whether the perturbation causes a calving cliff thicker than flotation or a floating ice shelf to

10   form.

Maintaining constant ice thickness at the calving front while bedrock depth changes has a significant effect on the extensional stress at the grounding line. Consider the case of a grounded calving front when $H'_f < 0$; the grounding line thickness is the

calving front thickness $H_g = H_c \approx H_{f0}$. The stress condition (9d) can also be approximated to first order in $H'_f$ as

$$\Sigma_g = 4|U_X|^{1/n-1}U_X \approx (1-r)H_{f0}/2 - H'_f \qquad \text{at } X = 0. \tag{15}$$

As we have assumed that the flotation thickness perturbation $H'_f$ is negative, we see an increase in extensional stress relative to the unperturbed flotation thickness $H_{f0}$. Even for small $H'_f$, the increase can be very significant: with $1 - r = 0.1$, $H'_f$ only

needs to be 1/20 of the unperturbed flotation thickness $H_{f0}$ in order for the stress perturbation to be the same size as the unperturbed stress $(1-r)H_{f0}/2$. An illustration of this effect is given in panel a of Fig. 8, where a slight decrease in flotation thickness (panel a1) clearly leads to a substantial increase in extensional stress (panel a2).

    The extensional stress perturbation occurs because the calving cliff ice thickness has not changed at first order, but bedrock depth is shallower. The calving cliff now protrudes further above the water line and the depth-averaged normal stress exerted

on it by the water is smaller. As a result, the extensional stress in the ice has to increase. This increase in stress is what leads to the increase in flux caused by the decrease in flotation thickness $H_f$. In fact, for small $H'_f < 0$, (11) then becomes

$$Q \approx \left(\frac{1-r}{8}\right)^{n^2/(n+1)} \frac{H_{f0}^{(3n+1)/(n+1)}\left[H_{f0} - 2(1-r)^{-1}H'_f\right]^{n^2/(n+1)}}{\left(\Gamma + W^{-(n+1)/n}H_{f0}\right)^{n/(n+1)}} \tag{16}$$

and $Q$ increases as $H'_f$ becomes more negative.

    This is consistent with the behaviour shown in Fig. 5. For grounded calving, we always find the anomalous relationship

between $Q$ and $H_f$, regardless of the basal friction parameter. Panels b and c of Fig. 7 also show that (16) is accurate only for very small $H'_f$; this is presumably a result of the fact that $1 - r = 0.1$ is not extremely small, and of the fact that the quadratic term in (14) starts to become large enough to affect results (indeed, panel a of Fig. 2 indicates that a linearization of $\phi$ is unlikely to be accurate for grounded calving fronts except very close to $H_f/\Lambda = 2$.)

## 5.3   Floating calving fronts

We can conversely take the case of $H'_f > 0$, which leads to the formation of a floating ice shelf. As the calving front thickness does not change to first order, the extensional stress at the calving front remains equal to

$$\Sigma(X_c) = 4|U_X|^{1/n-1}U_X = (1-r)H_{f0}/2 \tag{17}$$

Suppose that basal drag is not so large as to render lateral drag insignificant on the grounded portion of the boundary layer. In that case, even though the ice at the grounding line is slightly thicker than at the calving front (by $H'_f$), the driving stress

in the floating ice shelf is small compared with the other forces acting on the shelf. In particular, the surface slope of the ice shelf is small because of the small density difference between ice and water. Most of the reduction in thickness between grounding line and calving front is accounted for by the bottom of the ice shelf sloping upwards (see panel b1 of Fig. 8). The surface slope of the ice shelf (which causes the driving stress) is only $(1-r)/r$ times the bottom slope. Since the driving stress is weak, the dominant balance of forces on the ice shelf is then between the gradient of depth-integrated extensional stress

$(H\Sigma)_X = 4(H|U_X|^{1/n-1}U_X)_X$ and the lateral drag $W^{-(n+1)/n}H|U|^{1/n-1}U$: in other words, (9a) becomes approximately

$$(H\Sigma)_X - W^{-(n+1)/n}H|U|^{1/n-1}U \approx 0, \tag{18}$$

with the driving stress an $O(1-r)$ correction. It follows that the floating ice shelf acts to reduce extensional stress at the grounding line relative to its value at the calving front; this is the 'buttressing' effect of the ice shelf.

For small $H'_f$, we obtain a short ice shelf and can treat $H \approx H_{f0}$ and $U \approx Q/H_{f0}$ in the shelf as constant, so that

$$\Sigma(X) \approx (1-r)H_{f0}/2 - W^{-(n+1)/n}Q^{1/n}H_{f0}^{-1/n}(X_c - X),$$

$X_c$ being the length of the floating ice shelf. This effect — the linear reduction in extensional stress in the floating shelf with distance from the calving front — is illustrated in panel b of Fig. 8.

The shelf length is dictated by $H'_f$. A larger flotation thickness requires a longer ice shelf before the calving front thickness $H_{f0}$ is reached, potentially leading to more buttressing. The ice shelf thickness gradient is $-H_X = HU_X/U \approx H_{f0}^2\Sigma^n/(4^nQ)$, and the shelf length $X_c$ is constrained by the fact that the decrease in ice thickness between grounding line and calving front is $H'_f \approx \int_0^{X_c} -H_X \mathrm{d}X$. This allows us to compute $X_c$ and hence the stress at the grounding line

$$\Sigma_g = \Sigma(0) \approx \left\{ \left[ \frac{(1-r)H_{f0}}{2} \right]^{n+1} - \frac{(n+1)4^nQ^{(n+1)/n}H'_f}{W^{(n+1)/n}H_{f0}^{(2n+1)/n}} \right\}^{1/(n+1)}. \tag{19}$$

Substituting this in (11), flux satisfies for small $H'_f > 0$,

$$Q \approx \frac{H_{f0}^{(3n+1)/(n+1)}}{\left( \Gamma + W^{-(n+1)/n}H_{f0} \right)^{n/(n+1)}} \left\{ \left[ \frac{(1-r)H_{f0}}{8} \right]^{n+1} - \frac{(n+1)Q^{(n+1)/n}H'_f}{W^{(n+1)/n}H_{f0}^{(2n+1)/n}} \right\}^{n^2/(n+1)^2} \tag{20}$$

At first glance, it does not seem that (20) is much use — it defines $Q$ implicitly. However, from (19), it is not difficult to show that an increase in $H'_f$ leads to a decrease in extensional stress $\Sigma_g$ at the grounding line and, therefore, to a decrease in flux $Q$. The stress decreases because the ice shelf lengthens as $H'_f$ increases and the total amount of lateral drag on the ice shelf increases.

Again, we have given an *ad hoc* derivation for (20). We can formalize that derivation as shown in the supplementary material, once more based on the small density difference $1-r$. Panel b of Fig. 7 shows that (20) is more qualitatively than quantitatively accurate for the case of no basal friction $\Gamma = 0$. Again, this is presumably the result of $1-r$ not being extremely small, and of higher order terms in the approximation scheme above becoming important.

However, as panel a of Fig. 7 also shows, the anomalous behaviour disappears entirely for floating ice shelves when the basal friction coefficient $\Gamma$ becomes large. In such cases, the argument above must become qualitatively incorrect. The change-over from the anomalous behaviour for flow dominated by lateral drag to the 'normal' behaviour obtained with significant basal drag occurs because when the basal friction coefficient is large, ice velocities near the grounding line become small. This has two effects: it (i) reduces the lateral drag term $W^{-(n+1)/n}H|U|^{1/n-1}U$ in (9a) and (ii) increases the thickness gradient and, therefore, the driving stress. Specifically, conservation of mass in the floating shelf dictates that

$$H_X = -HU_X/U = -H|\Sigma|^{n-1}\Sigma/(4^nU), \tag{21}$$

so that $H_X$ becomes large when $U$ is small. The driving stress $-(1-r)HH_X$ can then no longer be ignored in (9a); (18) is no longer applicable, and neither is (20). An increase in flotation thickness can now potentially cause an increase in ice flux, at least when the calving front is afloat. This is shown in panel c of Fig. 8, and is also described in more formal detail in the supplementary material.

5      It is relatively straightforward to estimate how large $\Gamma$ needs to be in order for driving stress to appear at leading order in the shelf. Note that thickness, velocity, and stress are continuous across the grounding line. As a result of (21), so is the thickness gradient $H_X$, but not the surface slope. With large $\Gamma$ on the grounded side, driving stress balances basal shear, $-HH_X \sim \Gamma|U|^{1/n-1}U$. In order for driving stress to appear at leading order in the shelf, it should be comparable to lateral drag, so $-(1-r)HH_X \sim W^{-(n+1)/n}H|U|^{1/n-1}U$. It follows that

$$\Gamma W^{(n+1)/n}/H \sim \Gamma W^{(n+1)/n}/(2\Lambda) \sim (1-r)^{-1}. \tag{22}$$

With $r = 0.9$, this corresponds to $\Gamma W^{(n+1)/n}/\Lambda \sim 20$, consistent with panel a in Fig. 7, where the green line corresponds to $\Gamma W^{(n+1)/n}/\Lambda = 25$.

     Finally, consider the limiting case of very large basal friction coefficient (meaning, $\Gamma W^{(n+1)/n}/H \gg (1-r)^{-1}$) combined with an ice shelf that has limited extent. By an extension of the argument above, this corresponds to a large driving stress and to lateral drag playing an insignificant role in force balance. In this case, we can make our theory agree with previous work for laterally unconfined flow in Schoof (2007a) by simply ignoring lateral drag in (9a),

$$(H\Sigma)_X - (1-r)HH_X = 0.$$

Integrating and applying the boundary condition (17) shows that extensional stress at the grounding line is simply given by

$$\Sigma_g = (1-r)H_f/2.$$

This is the same extensional stress at the grounding line as we would expect in the case of the FL model. The flux increases monotonically with floatation thickness when this is substituted into (11) and $\Gamma$ is assumed large,

$$Q \approx \left(\frac{1-r}{8}\right)^{n^2/(n+1)} \frac{H_f^{(n^2+3n+1)/(n+1)}}{\left(\Gamma + W^{-(n+1)/n}H_f\right)^{n/(n+1)}}$$
$$\approx \left(\frac{1-r}{8}\right)^{n^2/(n+1)} \frac{H_f^{(n^2+3n+1)/(n+1)}}{\Gamma^{n/(n+1)}}. \tag{23}$$

Since we are assuming that $m = 1/n$, this is actually nothing more than a scaled version of equation (3.51) in Schoof (2007a). As panel a of Fig. 7 shows, we do get agreement between the CD calving law results and the FL model for large basal friction coefficients, at least while $H_f/\Lambda$ remains close enough to the critical value of 2: the flux curves then agree well with each (as

20 indicated by the arrow labelled 'driving stress dominant').

     For larger $H_f/\Lambda$, this agreement ceases. The shelf gets long enough that, even with large enough basal friction on the grounded portion, lateral drag on the floating shelf cannot be ignored. The next section describes in more detail the mechanics of a very long ice shelf.

## 5.4 The finite flux limit for large flotation thickness

For a fixed value of $\Lambda$ and large flotation thickness $H_f$, the flux $Q$ appears to approach a finite limit in panel a of Fig. 5. That limit is of the form

$$Q \sim (1-r)^n C(n) W (2\Lambda)^{n+1}, \tag{24}$$

with $C \approx 8.67 \times 10^{-3}$ for $n = 3$, regardless of the choice of basal drag parameter $\Gamma$. The physics behind this is relatively simple: a large value of $H_f$ corresponds to a large difference between ice thickness at the grounding line and at the calving front, which in turn requires a long ice shelf. With a long ice shelf, most of the floating ice shelf becomes fully 'buttressed,' in the sense that extensional stress gradients are weak and there is a balance between driving stress and lateral drag as well as basal drag in the grounded part of the glacier. In other words, a local force balance persists around the grounding line, and extensional stresses only become significant in the floating shelf close to the calving front. Moreover, since we are assuming that the floating shelf is still short compared with the length of the glacier and do not include basal melting in our model, ice flux also varies insignificantly along the floating shelf. Hence, the flux through the grounding line is the same as the flux through the calving front. Importantly, we now have a situation in which ice flux through the grounding line is determined entirely by the calving parameter $\Lambda$, and independent of depth to bedrock at the grounding line.

This situation was previously explored by Hindmarsh (2012) and Pegler (2016). These authors find that ice flux through the calving front is determined in a boundary layer around the calving front in which extensional stress is significant. In our notation, the boundary layer takes exactly the same form as (9) for floating ice ($\theta = 0$), but with different matching conditions:

$$4(H|U_X|^{1/n-1}U_X)_X - W^{-(n+1)/n}H|U|^{1/n-1}U - (1-r)HH_X = 0 \tag{25a}$$

$$(UH)_X = 0 \tag{25b}$$

for $X < X_c$, with

$$4H|U_X|^{1/n-1}U_X = (1-r)H^2/2, \qquad H = H_c \qquad \text{at } X = X_c \tag{25c}$$

where $H_c$ is the prescribed calving front thickness, and

$$UH \to Q, \qquad W^{-1/n-1}Q|U|^{1/n-1} \sim -(1-r)(Q/U)(Q/U)_X,$$

$$U \to 0 \qquad \text{as} \qquad X \to -\infty. \tag{25d}$$

The analysis of this boundary layer (a formal derivation of which is included in the supplementary material) is much the same as for (9), and $Q$ satisfies a power law relationship with ice thickness $H_c$ and channel width $W$ at the calving front, of the form $Q \propto (1-r)^n C(n) W H_c^{n+1}$. In the CD model, the ice thickness at a floating calving front is $H_c = 2\Lambda$, which gives the flux

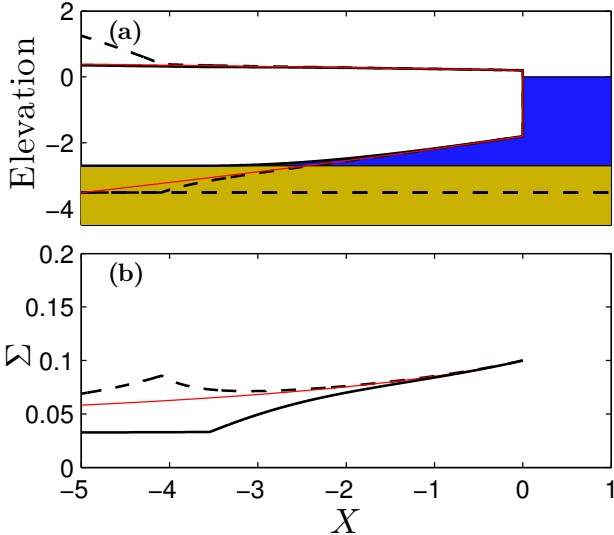

**Figure 9.** Solutions for a long shelf, same plotting scheme and parameter values as in Fig. 8. The dashed line shows the solution with $\Gamma = 125$, $H_f = 3.9$, the solid line shows $\Gamma = 0$, $H_f = 3$. Both are marked with red diamonds in panel a of Fig. 7. The red line shows the solution to the near-calving-front boundary layer (25). All three solutions agree closely with this boundary layer near the calving front.

relationship (24). A more detailed derivation of the relationship (24) is given in the supplementary material, and a numerical value computed from the reduced boundary layer model presented there agrees very well with that given above, obtained from the solutions to the full boundary layer problem (9) for large values of $H_f/\Lambda$. In fact, convergence to that value is very rapid as $H_f/\Lambda$ increases, as shown in panel a of Fig. 7. Fig. 9 shows that for even moderately large values of $H_f/\Lambda$ (so when the

ice shelf still has relatively limited extent), the thickness and stress profile near the calving front is well-approximated by the solution to (25), regardless of the amount of basal friction in the grounded part of the glacier. The fluxes in all three examples shown in Fig. 9 are almost identical.

Consider the special case of no basal drag on the grounded part of the glacier. We can show how (24) confirms that we expect an anomalous flux-depth-to-bedrock relationship due to buttressing in the ice shelf. Take a grounded calving cliff just

at flotation, with thickness $H_f = H_c$. The flux is given by (12) with $\Gamma = 0$. Compare this with the flux through a long floating ice shelf that terminates in a calving cliff of the same thickness $H_c$. The solution (24) then predicts that the flux through the floating shelf is smaller than through the grounded calving front, even though the two have the same thickness. This is true at least when the density difference $1 - r$ is small, because the exponent $n^2/(n+1)$ on $(1-r)$ in the formula for flux for the grounded cliff in (13) is smaller than the exponent $n$ for the floating cliff in (24). This provides further evidence for the

buttressing action of the ice shelf leading to an anomalous flux-flotation-thickness relationship.

## 6   Discussion and conclusions

In this paper, we have applied the boundary layer analysis of Schoof (2007a) to a model for channelized outlet glacier flow, incorporating a parameterised description of lateral drag (Dupont and Alley, 2005) and a simple calving law due to Nick et al. (2010). The purpose of this work has been to show how calving and lateral drag can potentially combine to produce a very different relationship between ice flux at the grounding line and glacier bed geometry from that for laterally unconfined marine ice sheet flow. For the latter, ice flux is an increasing function of depth to bedrock, while for a channelized outlet glacier, we find that an 'anomalous' relationship in which flux decreases with increasing depth to bedrock is possible (panel a of Fig. 5 above).

Such an anomalous relationship has significant consequences for stable glacier margin positions. Consider the model (8) for the flow of the glacier as a whole. Two boundary conditions apply at the free boundary $x = x_g$. One of these is a thickness condition, while the second is the flux condition (10a), which can be written in the form

$$q(x_g(t),t) = Q_g\left(-b(x_g(t)), w(x_g(t)), \lambda, \gamma, \varepsilon, m, n, r\right),$$

where $q = -w^{n+1}h^{n+1}(h + w^{(n+1)/n}\gamma)^{-n}|(h+b)_x|^{n-1}(h+b)_x$ is ice flux, and $Q_g$ is the flux $Q$ predicted by the boundary layer problem, written in terms of the original dimensionless parameters and the channel geometry at the grounding line.

Steady states can now be computed easily from (8). To determine their stability, the theory of Fowler (2011) and Schoof (2012) can be extended straightforwardly to the present case, the only modification required being the generalization of the thickness variable $h$ and flux variable $Q$ in Schoof (2012) to our $wh$ and $wq$, respectively. It then follows that a steady state is linearly stable if and only if (see e.g. condition (5.1) in Schoof (2012))

$$\frac{\mathrm{d}}{\mathrm{d}x_g}\left[w(x_g)Q_g\left(-b(x_g), w(x_g), \lambda, \gamma, m, n, r\right)\right] > w(x_g)a(x_g) \tag{26}$$

If $Q_g$ does not necessarily increase with $b$, then steady grounding lines located on a downward-sloping beds can become unstable. This is illustrated in Fig. 3, where steady grounding line positions on a downward-sloping bed are plotted against the accumulation rate $a$ over the ice sheet, which is assumed to be spatially uniform, as is channel width $w$. The steady state grounding line position of (8) is defined implicitly by

$$wQ\left(-b(x_g), w, \lambda, \gamma, m, n, r\right) = awx_g.$$

Treating $x_g$ as a function of $a$ and differentiating both sides with respect to $a$, we have

$$\left[\frac{\mathrm{d}}{\mathrm{d}x_g}\left[wQ_g\left(-b(x_g), w(x_g), \lambda, \gamma, m, n, r\right)\right] - aw\right]\frac{\mathrm{d}x_g}{\mathrm{d}a} = wx_g,$$

and when the stability condition (26) is satisfied, $\mathrm{d}x_g/\mathrm{d}a > 0$, so that a stable grounding line must advance when $a$ is increased: a stable ice sheet gets larger when it receives more surface snowfall. Fig. 3 shows a solution branch for which this is not the case, even though the grounding line is located on a downward-sloping bed.

Conversely, we may see grounding lines attain stable steady state positions on upward-sloping beds if $Q_g$ decreases with depth to bedrock $-b$: Fig. 4 shows several examples in which the steady state grounding line advances up a reverse bed slope

as accumulation rates are increased. A second mechanism by which such stabilization on upward-sloping beds can occur is the dependence of discharge $wQ_g$ on width $w$: a sufficiently narrow bottleneck in the channel could stabilize a grounding line on an upward slope even if $Q_g$ did increase with depth $-b$, because $wQ_g$ is an increasing function of $w$ (this argument is due to Jamieson et al., 2012). This second mechanism is however not responsible for the behaviour shown in Fig. 4, where channel width is constant along the domain. It is worth noting that simulations of Greenland outlet glaciers using the CD calving law (Nick et al., 2010) have similarly produced steady states located on upward-sloping beds. Our work suggests that this may be due not only to narrowing of the channel but also to the calving law.

Our aim has not been to be authoritative in establishing the existence of an anomalous flux-depth relationship: our model contains at least three components that can be improved upon. First, the parameterised description of lateral drag should eventually be dispensed with, replacing our model with one that resolves the cross-channel dimension. The scaling that underlies our boundary layer model should still be applicable in that case, but the actual boundary layer model will consist of a set of coupled partial differential equations (as opposed to ordinary differential equations) and is likely to be much more onerous to solve for a large number of parameter combinations, as we have been able to do here.

Second, we have neglected the effect of basal melting on the shelf here. This is tractable in the framework we have developed here with a simple, prescribed basal melt rate, but doing so still introduces sufficient complications to lie beyond the scope of a single paper; a second manuscript that incorporates melting into our theory is in preparation.

Third, the calving law we have employed is relatively poorly constrained by observation and is based on a number of simple assumptions about how cracks form near a calving front. Furthermore, it relies entirely on water depth in surface crevasses as a control parameter that should itself be determined by additional physics governing the drainage of surface melt water. We have chosen to take the calving model at face value, simply prescribing the crevasse water depth as a control parameter. This is worth emphasizing as the dependence of calving cliff height on flotation thickness predicted by the calving model turns out to be key to the anomalous flux-depth relationship. It is likely that other, more sophisticated calving models (for instance one based on the formulation in van der Veen (1998a, b)) can also be written in the form of a calving cliff height as a function of crevasse water depth, though presumably with a different specific from the CD model: as in the latter, surface hydrology becomes a key component in understanding calving.

For a floating ice shelf, calving cliff height in the CD model is simply proportional to crevasse water depth and independent of depth to bedrock. In other words, the CD model can then be thought of as a generic calving model that imposes a fixed thickness at the floating glacier terminus. Moving the grounding line to a location with greater flotation thickness (or equivalently, depth to bedrock) therefore leads to a longer ice shelf forming before it can reach the prescribed calving cliff height. If the mechanical effect of the ice shelf is primarily to provide lateral drag, then a longer shelf leads to a greater reduction in extensional stress between calving front and grounding line, and therefore to lower ice flux despite a greater depth to bedrock at the grounding line. Whether this occurs or not is a function of basal drag on the grounded part of the glacier: if basal drag upstream of the grounding line is moderate compared with lateral drag, then the surface slope and driving stress of the floating shelf will be small, so the effect of the shelf is mostly to generate lateral drag. By contrast, if basal drag is large upstream of the grounding line, then the floating shelf will be relatively steeply sloped and lateral drag will play a lesser role in force balance there, leading

to the possibility that the floating shelf does not cause a reduction in extensional stress and hence flux through the grounding line. The changeover between the two regimes happens when, in the notation of section 3.1, the basal drag coefficient is approximately (see equation (22) above)

$$\gamma \sim (1-r)^{-1} w^{-(n+1)/n} b,$$

and $\lambda \sim b$.

As we have indicated, the thickness of floating calving fronts in the CD model is uniquely controlled by the crevasse water depth parameter, and does not depend on depth to bedrock. The same generic relationship between ice flux and depth to bedrock at the grounding line will therefore be obtained for any other calving law that fixes the height of a floating calving front independently of depth to bedrock. By contrast, the CD model results are unlikely to be robust in the same way for grounded calving fronts. Specifically, for a grounded calving front, the Nick et al. (2010) calving law predicts that calving cliff height decreases relatively slowly when the calving front is moved to a location with shallower depth to bedrock. In turn, this leads to more of the calving cliff being exposed above the water line, and consequently to larger extensional stresses acting on the calving front, and these larger extensional stresses cause ice flux to increase as depth to bedrock is decreased.

This contrasts with an alternative 'calving at flotation' (FL) calving law, in which calving front height is always proportional to depth to bedrock and no floating shelf forms. In that case, extensional stress at the grounding line increases with depth to bedrock, and so does ice flux.

We close by noting that our approach can potentially be used to study the effect of other calving laws relatively simply in future, by replacing the function that specifies ice thickness at the calving front. Since an anomalous flux-depth-to-bedrock relationship may be possible and would have significant consequences for stable outlet glacier configurations, and it may be worth testing this before embarking on simulations of actual glaciers using different calving laws.

## 7  Code availability

The MATLAB code used in the computations reported is included in the supplementary material.

## Appendix A:  A note on direct solutions of the steady state problem

At issue is the uniqueness of the orbit that emerges from the fixed point of the dynamical system (6), at which $(h, \sigma, q, \chi) = (h_0, [a(0)/h_0]^{1/n}, 0, 0)$: only when uniqueness is guaranteed does the shooting method of section 3.2 make sense. Linearising the dynamical system around the fixed point leads to a problem with eigenvalues $0$ (repeated), $a(0)w(0) > 0$ and

$$-[4\epsilon^{-1} n h_0 (a(0)/h_0)^{1/n} - (n-1)]a(0)w(0).$$

The sign of this last eigenvalue is negative when $\epsilon$ is small enough. In that case, the fixed point has a two-dimensional centre manifold, a stable and an unstable manifold (Wiggins, 2003). The centre manifold has no dynamics (it consists of other fixed

points, corresponding to different values of $h_0$, or orbits that do not satisfy $\lim_{\eta \to -\infty} \chi = 0$), so that the fixed point must be approached in the limit $\eta \to -\infty$ along the unique unstable manifold.

## Appendix B: The boundary layer problem

The simplified forms of the flux law in (10) can be derived by a transformation of the boundary layer problem (9), using

$$U = Q^{n/(n+1)}W^{1/(n+1)}\mathcal{U}, \qquad H = Q^{1/(n+1)}W^{-1/(n+1)}\mathcal{H}, \qquad X = W\mathcal{X} \tag{B1}$$

and

$$\mathcal{C} = \Gamma W^{(m+n+3)/(n+1)}/Q^{(2-nm)/(n+1)},$$

$$\mathcal{H}_f = H_f(W/Q)^{1/(n+1)}, \qquad \mathcal{L} = \Lambda(W/Q)^{1/(n+1)}, \qquad X_c = W\mathcal{X}_c. \tag{B2}$$

With these definitions, it is easy to show that (9) is invariant under the transformation

$$(U, H, X, X_c, H_g, W, Q, \Gamma, \Lambda) \mapsto (\mathcal{U}, \mathcal{H}, \mathcal{X}, \mathcal{X}_c, \mathcal{H}_g, 1, 1, \mathcal{C}, \mathcal{L}).$$

The parameters in this re-scaled version of the model are $r$, $n$, $m$, $\mathcal{H}_f$, $\mathcal{C}$ and $\mathcal{L}$, while $\mathcal{U}$, $\mathcal{H}$ and $\mathcal{X}$ are dependent and independent variables, respectively. It then can be shown that the transformed boundary layer problem has a solution if and only if the parameters $r$, $n$, $m$, $\mathcal{H}_f$, $\mathcal{C}$ and $\mathcal{L}$ satisfy some functional relationship with each other. Using this fact, it is easy to show that the simplified flux laws (10) must hold.

Deriving that functional relationship between $r$, $n$, $m$, $\mathcal{H}_f$, $\mathcal{C}$ and $\mathcal{L}$ can be done by a further coordinate transform of the dependent variables (see also appendix A in Schoof (2011))

$$\mathcal{Q} = \mathcal{U}\mathcal{H}, \qquad \Psi = \mathcal{Q}^{-1}\mathcal{U}^{-(2n+1)/n^2}|\mathcal{U}_{\mathcal{X}}|^{1/n-1}\mathcal{U}_{\mathcal{X}}, \qquad \xi = \mathcal{U}^{(n+1)^2/n^2}. \tag{B3}$$

and switching to an independent variable $\zeta$ defined through

$$\zeta = \int_0^{\mathcal{X}} \mathcal{Q}(\mathcal{X}')^{-1}\Psi(\mathcal{X}')^n \xi(\mathcal{X}')^{-1/(n+1)}\mathrm{d}\mathcal{X}',$$

This transforms the boundary layer problem into a non-singular dynamical system in which the matching conditions (9g) correspond to a fixed point $(\Psi, \xi, \mathcal{Q}) = (1, 0, 1)$ being attained as $\zeta \to -\infty$, and there is a unique orbit along which this can happen; to prove the uniqueness of that orbit, an additional transformation to $\nu = \xi^{(n^2(m+1)-n)/(2(n+1)^2)}$ may be required, but the basic argument remains the same as in Schoof (2011). The boundary conditions (9d)–(9e) then provide two further constraints: one on the location of the point along the orbit that corresponds to the calving front, and another to relate the parameters of the model, leading to a functional relationship between $r$, $n$, $m$, $\mathcal{H}_f$, $\mathcal{C}$ and $\mathcal{L}$. In practice, this must be solved

for numerically by integrating the transformed dynamical system. More complete details on the solution of the boundary layer problem can be found in the supplementary text and the numerical code provided as supplementary material for this paper.

*Acknowledgements.* CGS acknowledges the support of NSERC grants 357193-13 and 446042-13. ADD's work is supported by the U.S. Department of Energy, Office of Science, Office of Advanced Scientific Computing Research, under award number DE-SC0010518. VTP was supported by the NSERC CREATE AAP program. We are grateful to the three anonymous referees and the editor for their thorough scrutiny.

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
