# Peer review of "Boundary layer models for calving marine outlet glaciers"

_The Cryosphere, 2017_

## Referee Comment (RC1) · Anonymous Referee #1 · 10 May 2017

After the boundary layer model for marine ice sheets, Christian Schoof and co-authors present in this paper an extension of the boundary layer model by investigating two calving laws, i.e., the CD model due to Nick et al; Benn et al., and one simple model based on calving at flotation. They find that for unbuttressed ice sheets, the flux according to the CD model can decrease with increasing water depth at the grounding line, which is counterintuitive with respect to the stability criterion that was proposed in Schoof (2007). This is the first rigorous assessment of the widely used calving model and allows for a better comprehension of its behaviour. Furthermore, it suggests that the fact that steady-state grounding line positions can be obtained on retrograde slopes are not necessarily due to an inland narrowing of the channel, but also due to the calving law.

The paper is quite lengthy but gives (in my opinion) a rigorous mathematical derivation

of the models and the boundary layer model derived from it. However, while the authors repeatedly state that they take the CD model at face value, they should - when presenting that model - demonstrate in a comprehensive way what the pros and cons of the model are. Moreover, as the CD model (and derivatives) are widely used, some more criticism and lines for improvement are in order. In that respect, Figure 3 is quite enlightening showing that grounding line positions for the CD model show this rather non-intuitive characteristic (as a function of accumulation rate).

Another important contribution of this work is that it lays out a basis for further analysis of new calving laws by adapting the code available in the supplementary section (see discussion and conclusions). This should also be mentioned in the abstract.

Throughout the paper, the authors investigate the case of a downward-sloping bed (prograde slope). However, as shown in Schoof (2007), retrograde (upward-sloping) beds do not allow for steady-state grounding line positions in absence of buttressing. Gudmundsson et al (2013) demonstrated that stable steady states on such slopes may occur due to ice shelf buttressing. Also in Greenland, where the CD calving law has been mostly applied, retrograde slopes occur. Therefore, it would be interesting not to limit the analysis to downward sloping beds, but to investigate (albeit briefly) the behaviour of the CD calving law in regions with overdeepenings.

Detailed remarks

Page 2, line 6: grounding line

Page 3, line 3: commas between references

Page 2, line 7: assumed constant in time

Page 4, line 4: even when neither of the two limits

Page 4: line 5-7: Given that the use of this model is essential throughout the analysis, it would be good to bring in some more solid arguments in favour its use. Stating that the simplification works reasonable well and that you analyse the model at face value

is somehow weak.

Page 9: bottom equation: [x] instead of ]x]

Page 16, line 12: converges to the one (or state flux conditions instead)

Page 21, line 7: bigger -> larger

Page 22, line 18: formula -> eq.
* * *

---

## Referee Comment (RC2) · Anonymous Referee #2 · 15 May 2017

This manuscript extends the boundary layer theory of Schoof (2007) to the case for calving marine outlet glaciers. Lateral drag from sidewalls provides another constraint on stable grounding line positions for confined outlet glaciers; the authors show that steady grounding line positions can be obtained on upward sloping beds with this effect considered. Since the length of any floating ice shelf affects the amount of lateral drag, the authors identify calving as an important mechanism and derive grounding line flux as a function of relevant calving parameters for two calving laws. These are the CD model and one for calving at flotation.

The work presented is an extension of Schoof (2007), taking the analytical model further. While it is already known that sidewall drag influences grounding line positions (e.g. numerical solutions by Gudmundsson) a boundary layer formulation such as this one has not been written down before and this paper therefore provides a worthwhile

extension to previous work. Furthermore, the authors provide the code, which will allow others to analyse effects with different calving laws.

I have not gone through every mathematical detail, but it certainly appears that the authors have been thorough in their approach. Most of the corrections I suggest below are relatively minor. The main thing I would like is a bit more discussion surrounding the background to the calving laws and why these ones were chosen. Perhaps introduce a subsection into the model section describing the calving laws and their background in a bit more detail and in a general context. Furthermore, I think it would be useful to have some discussion towards the end of the paper about how, qualitatively, you expect processes to be affected by e.g. different choice of calving laws, which include different mechanisms, and basal melt (exclusion of it mentioned line 14, section 2). Perhaps insert a separate discussion and conclusion. This would help make the paper more accessible to a general reader who is interested in what the key parameters really are.

I would also really appreciate a table of variables being included. There were several points in the manuscript where this would have been useful to reference as so many different variables are used.

Minor comments

- **Abstract, line 3-4** Re-phrase as confusing ordering at the moment. I suggest 'The length of any floating ice shelf present also affects the lateral drag, hence calving is an important process'.

- **Abstract, line 9** 'increasing depth to bedrock' - better at this point to refer to as 'retrograde bed slope' as this how people usually think of it?

- **Intro, line 19-23** Sentence far too long. Insert full stop after first part. i.e. '...that can alter the flux-to-bedrock-depth relationship. These include...'

- **Intro, line 15 onwards** Can you insert a sentence or two justifying choosing these two calving laws over others? Or say you'll do this in model description section and add discussion in there as mentioned above.

- **page 3, line 10** Why using B, rather than more standard choice of A for Glen coefficient?

- **page 3, line 24** Can you extend discussion here with a couple of sentences about limitations of this parameterisation?

- **Figure 1** Table of variables would certainly help reader when looking at this figure. Also, for a grounded terminus shouldn't inequality for $h$ actually read $h_f \leq h_c$ as still grounded if calving happens *at* flotation. Alternatively, perhaps you could insert a third lower diagram illustrating $h_f = h_c = h_g$ since this could then be used as a reference when describing the second calving law you use (page 5, lines 1-2)?

- **Figure 2** Grey shaded regions do not show at all when printed. Make darker.

- **Figure 2** Insert space 'Panel (b)' in caption

- **equation 1h** I think this is the first time $d_w$ is used but you do not explicitly state that it is water depth. At least have in table.

- **equations 1h/1i** Can you line these up properly so 'at and if' are in line (and do similarly at several other points in paper).

- **page 7, lines 1-12** As mentioned above I think you want more discussion and context here. Given the length of this section I think it would also be helpful to split section 2 up into a couple of subsections e.g. 'ice flow model', 'calving laws'. This would help remind reader where to reference back to when thinking about the different parameterisations later.

- **section 3.1** a few more words reminding us what equations are e.g. page 8 line 5 'if $rh \geq b$ i.e. flotation.

- **page 9, line 9** '...glacier terminus, *which are* of the form'

- **page 9, line 14** change to 'monotonically downward'

- **page 11, line 24ish** confusing now having 'B' as a rescaled b and having the $\overline{B}$ etc earlier for Glen. Preferably change Glen to A but at least introduce table.

- **page 12, line 27** Physical interpretation of $\lambda$ being small?

- **page 13, line 4-8** Long sentence, difficult to take in. Split up.

- **page 14, line 1-2** 'on the then-redundant parameter $\Lambda$ to write alternatively' -> 'on the now-redundant parameter $\Lambda$ to simplify the expression to'

- **page 14, line 25** 'the CD model produces the same result as..., which is reassuring'.

- **page 17, eqn 11** insert fullstop.

- **page 22, line 13ish** Sentence between the two equations (line numbering gone askew here) should read 'Integrating and applying the boundary condition shows that extensional stress...'

- **page 22, eqn 23** insert fullstop.

- **page 25, line 16** You seem to sometimes talk in terms of the rescaled $B$ (like here) and at other points in terms of depth to bedrock $b$ (e.g. line 20). I would stick to the variable $b$?

Interactive
comment

[Figure]

- **page 26, line 16 onwards** Yes, this and the following discussion are good points but then the paper ends rather suddenly. Could you put this into a bit more context and suggest extensions/alternative approaches.

- **page 27, line 8** 'may be possible at least in principle' - awkward wording.

- **page 27, line 12** 'At issue' change to'An important issue'

- **page 28, line 13** 'meaning a functional relationship' change to 'giving a functional relationship'
* * *

---

## Referee Comment (RC3) · Anonymous Referee #3 · 6 Jun 2017

The manuscript applies some fairly sophisticated analysis to a simple but widely applied model of a laterally confined glacier subject to one of two calving rules: a 'calving at flotation' model and a crevasse depth calving model (CD) a la Nick 2010. CD turns out to be turns out to be essentially a minimum thickness criterion because of the relationship between calving front thickness, stress and crevasse depth. I think this is fairly well known, but this paper makes good use of that fact to write down an expression for the free boundary condition. The problem is first examined with an unusual (but well described) numerical model, and then with an a analytic boundary layer model, similar to the well known Schoof 2007 model for laterally unconfined flow. Both models feature an interesting result for the CD model: a flux which can decrease with bedrock depth at the grounding line, which in turn allows for stable equilibria on retrograde slopes

and the like. Note that this mechanism is not (quite) the same as in e.g Gudmundsson 2012, but rather arises from the interaction between the calving model and the stress balance, e.g flux increases when bedrock depth decreases for a grounded terminus because the first order change is a taller cliff and greater extensional stress.

Overall this is a very strong paper, both in terms of the analysis and the explanation. I'm sure it will be well cited.

**Minor comments**

If the Nick et al crevasse depth calving model is ro be referred to as 'CD' (as in Nick et al 2010 IIRC) then why not have a similar acronym for the other model (I think Nick called it 'FL'?)

P5-6, eqns 1h,1i: Although the derivations of these expression is in the supplement, would it not make sense to say at this point that the arise from combining the relationship between stress and thickness at the calving front and the relationship between 'dry' crevasse depth and stress.

P6, L15: I don't dispute that sensitivity to $d_w$ (and the requirement that $d_w \sim h$) is a problem for CD: in fact I would go further and say it that in current applications it might be standing in for physics that has nothing to do with hydrology at all.

P9, L24: 'Our aim in what follows. . . ' rather than a single sentence, it might be helpful to quickly sketch out the line of thought. It was not until about P15 that I got the sense of that.

P12,L15: 'Despite working at leading order in $\epsilon$ we have retained two terms that contain factors of $\epsilon$ in (9) '. Slightly odd phrasing, which might give the impression that the terms are retained even though they are $< O(1)$ ? Both factors are (for the case $n-3, m=1/3$) $\epsilon^{-9/16}$ so $> O(1)$. In the next paragraph that requires $\gamma$ and $\lambda < O(1)$

for all terms to appear at the same order.

P13, eq 9g. Does the factor $|h_x|^1/m - 1$ arise in general? Or just because $m = 1/n$? Perhaps I missed a trick here, but if $\gamma$ and $b_x$ are small in (7a), the flux expression just depends on the wall drag and driving stress (so m does not enter).

P19: typo? $H'_{f0} - > H'_f$ in expressions above (14)

---

## Author Comment (AC1) · 25 Jul 2017

We would like to thank the referees for the detailed scrutiny of the paper. Below, we have directly addressed all comments that call for clarification or improvement of the paper, not those that describe the paper in more general terms.

The original comments are reproduced in bold face, our responses in plain text, with excerpts from the paper in quotation marks. New text is rendered in italics.

[Figure]

**1 Referee # 1**

- **However, while the authors repeatedly state that they take the CD model at face value, they should — when presenting that model — demonstrate in a comprehensive way what the pros and cons of the model are. Moreover, as the CD model (and derivatives) are widely used, some more criticism and lines for improvement are in order. In that respect, Figure 3 is quite enlightening showing that grounding line positions for the CD model show this rather non-intuitive characteristic (as a function of accumulation rate).**

We have tried to be as comprehensive here as is possible without trying to derive a more sophisticated calving model that incorporates the same basic physics as Nick et al (2010). We now include the following paragraphs in section 2.2; much of this material already appeared in the first draft, but the material highlighted in boldface below is new, adding to our discussion of the pros and cons of the model (italicized for new text)

"*In our view, the CD model is a cartoon version of the linear elastic fracture mechanics explored in by Weertman (1973,1980) and van der Veen (1998a,b). These papers consider the 'mode 1' (Zehnder, 2012) propagation of vertical cracks into ice under tensile (extensional) stresses. This is done by computing stress levels around the crack tip from known Green's functions for parallel-sided elastic slabs with cracks penetrating from the upper or lower surfaces, accounting for the pressure exerted by water in the cracks, and applying a fracture toughness criterion. The CD model by contrast assumes that extensional stress increases with depth in the ice in a linear, cryostatic fashion. The model then computes crevasse penetration as being the distance from the upper and lower surfaces at which that extensional stress becomes sufficiently negative (that is, sufficiently compressive) to overcome the pressure exerted by water at the same depth. The CD model therefore does not compute stress with the same level of sophistica-*
*tion as the papers by Weertman (1973,1980) and van der Veen (1998a,b), but follows the same basic approach of computing crevasse propagation based on a known ice geometry and known water pressures applied inside the crevasses, and it has the advantage of tractability.*

*The basic method in van der Veen (1998a,1998b) in principle allows for a constraint to be computed that links ice thickness, applied extensional stress, crevasse water level and fracture toughness at the moment that surface and basal crevasses together first penetrate through the entire ice thickness. Given that extensional stress is a function of ice thickness through (1e), this constraint could be converted into a criterion for the thickness $h_c$ at which calving occurs, giving a more sophisticated version of the Nick et al (2010) CD model. However, the papers by van der Veen do not deal with the case in which both, surface and basal crevasses are present and interact with each other (so the relevant Green's functions are not given), and he does not explicitly compute a condition for calving that could be put in the form (1f). As a result, we confine ourselves to the simpler CD model here.*

*One of the practical pitfalls of the CD model is that it predicts no calving at all if $d_w = 0$ and surface crevasses are free of water. It is possible that this is an artifact of the simple representation of stress in the CD model, where the tensile stress driving crevasse propagation is assumed to have the same dependence on depth below the ice surface regardless of whether a crevasse is present or not. In reality, the formation of crevasses that penetrate through a significant fraction of the ice shelf leads to extensional stress becoming more concentrated around the crack tips than for shallow crevasses (see for instance Fig. 4 of van der Veen (1998a)). This represents a positive feedback on crack propagation, and could lead to calving even for the case of water-free surface crevasses (see also Weertman, 1980).*

*More recently, others have extended the linear elastic fracture mechanics ap-*

*proach of Weertman (1973,1980) and van der Veen (1998a,b) to include effects such as the role distributed damage due to the formation of microcracks in initiating crevasse formation, the blunting of cracks tips due to viscous deformation, and the presence of significant torques near the calving front (Krug et al, 2014, Mobasher et al, 2016, Jimenez et al 2016, Hongju et al, 2017). The complexity of these processes however makes them difficult to parameterize in a model that does not resolve the scale of individual crevasses, and we do not consider them here.*

*The Nick et al (2010) CD calving model, along with the work of Weertman (1973,1980) and van der Veen (1998a,b), is based on tensile failure. We can contrast this with the shear failure model of Bassis and Walker (2011) (see also Bassis and Jacobs (2013) and Ma et al (2017)).* The CD model requires $d_w > 0$ and predicts calving for any $h$ below the value given by (1f), instantaneously removing all parts of the glacier shelf that are too thin. By contrast, the shear failure model of Bassis and Walker (2011) predicts that calving will start at a critical calving front thickness and not occur below that thickness, so the inequality in (1g) would need to be reversed. . . . "

- **Throughout the paper, the authors investigate the case of a downward-sloping bed (prograde slope). However, as shown in Schoof (2007), retrograde (upward-sloping) beds do not allow for steady-state grounding line positions in absence of buttressing. Gudmundsson et al (2013) demonstrated that stable steady states on such slopes may occur due to ice shelf buttressing. Also in Greenland, where the CD calving law has been mostly applied, retrograde slopes occur. Therefore, it would be interesting not to limit the analysis to downward sloping beds, but to investigate**

We have addressed this by adding a figure (Fig 4) to section 3.2, accompanied by the following text (again, italics indicate new text):

"*Fig. 4 shows analogous calculations to those in Fig. 3, but for an overdeepened bed shape based on that used in Schoof (2007b). For the FL model, we invariably see that an increase in accumulation rate makes the grounding line advance on a downward slope, and retreat on a retrograde slope. This is again analogous to the unbuttressed case studied in Schoof (2007b), where the grounding line is then unstable when located on an upward slope. For the CD model, the behaviour becomes more complicated. We see that the grounding line can either advance or retreat with increasing accumulation rate, on both the downward- and upward-sloping parts of the bed. Qualitatively, shallow water depths at the grounding line are more commonly associated with the standard, 'unbuttressed' behaviour (that is, an increase in accumulation tends to cause the grounding line to advance on downward slopes, and retreat on retrograde slopes). The reverse behaviour is associated with larger water depths at the grounding line. We also see that a decrease in $\lambda$ leads to the 'reverse' behaviour being observed down to shallower water depths at the grounding line, and in particular, through more of the overdeepened section.* "

In the conclusions section, we also refer back to this figure:

"Conversely, we may see grounding lines attain stable steady state positions on upward-sloping beds if $Q_g$ decreases with depth to bedrock $-b$: *Fig. 4 shows several examples in which the steady state grounding line advances up a reverse bed slope as accumulation rates are increased.* A second mechanism by which such stabilization on upward-sloping beds can occur is the dependence of discharge $wQ_g$ on width $w$: a sufficiently narrow bottleneck in the channel could stabilize a grounding line on an upward slope even if $Q_g$ did increase with depth $-b$, because $wQ_g$ is an increasing function of $w$ (this argument is due to Jamieson et al, 2012). *This second mechanism is however not responsible for the behaviour shown in Fig. 4, where channel width is constant along the domain.* It is worth noting that simulations of Greenland outlet glaciers using the CD calving

law (Nick et al, 2010) have similarly produced steady states located on upward-sloping beds. Our work suggests that this may be due not only to narrowing of the channel but also to the calving law."

- **Detailed remarks:**
  **Page 2, line 6: grounding line**
  **Page 3, line 3: commas between references**
  **Page 2, line 7: assumed constant in time**
  **Page 4, line 4: even when neither of the two limits**

  We have corrected all of these

- **Page 4: line 5-7: Given that the use of this model is essential throughout the analysis, it would be good to bring in some more solid arguments in favour its use. Stating that the simplification works reasonable well and that you analyse the model at face value is somehow weak.**

  We believe we are simply being honest here and throughout the paper by pointing this out directly. We describe our rationale for using the model at length in the introduction and the conclusions (namely, that it has been widely used elsewhere — so the ability to interpret existing results remains important — and is based on the physics involved, and furthermore, that the model allows rapid computation over large sets of parameter values); the passage flagged by the referee makes clear precisely what the downside of using the model is. The alternative would be to use a much more sophisticated and costly model; proceeding at "face value" is done in the hope that something *useful* can be learnt without resorting to that alternative, which would make it much more difficult to explore parameter space to the extent we are able to, and to come to *qualitative* conclusions. Of course, one could assert that only the best type of model should ever be used and that flow-line models are dead. We would respectfully disagree, and point to simple box models in glaciology and elsewhere as a tool still used to develop understanding.

Our effort should be seen as being in that spirit, but situated somewhere between box models and sophisticated three-dimensional multiphysics models.

To expand on where we address the rationale for the model, we have in the introduction

"We investigate how two particular calving laws that are relatively widely used in models for tidewater glaciers affect but- tressing in a simplified flowline model. The model lacks the sophistication of models that resolve the cross-channel dimension. Instead, it relies on a parameterization of lateral drag in terms of the mean along-channel velocity (Dupont and Alley, 2005; Nick et al., 2010; Jamieson et al., 2012; Hindmarsh, 2012; Pegler et al., 2013; Robel et al., 2014, 2016; Pegler, 2016). The chief advantages of the model are that it allows flux through the grounding line to be computed rapidly as a function of ice thick- ness through the use of a boundary layer theory (Schoof, 2007a) and that the role of different physical mechanisms becomes comparatively easy to trace. Future work will be required to address whether our results are reproduced qualitatively by more sophisticated (and more computationally intensive) models, and we hope that this paper can motivate such work."

and in the conclusions

"Our aim has not been to be authoritative in establishing the existence of an anomalous flux-depth relationship: our model contains at least two components that can be improved upon. First, the parameterised description of lateral drag should eventually be dispensed with, replacing our model with one that resolves the cross-channel dimension. The scaling that underlies our boundary layer model should still be applicable in that case, but the actual boundary layer model will consist of a set of coupled partial differential equations (as opposed to ordinary differential equations) and is likely to be much more onerous to solve for a large number of parameter combinations, as we have been able to do here."

Note that this is not new text, but hopefully addresses the point adequately.
- **Page 9: bottom equation:** $[x]$ **instead of** $]x]$
  **Page 16, line 12: converges to the one (or state flux conditions instead)**
  **Page 21, line 7: bigger → larger**
  **Page 22, line 18: formula → eq.**
  We have corrected these. For the page 16, line 12 correction, we state
  *As expected, the flux solutions obtained from the full steady state problem (6) for*
  *the CD and the calving at flotation models converge to those obtained from the*
  *boundary layer problem*

**2  Referee # 2**

- **The main thing I would like is a bit more discussion surrounding the background to the calving laws and why these ones were chosen. Perhaps introduce a subsection into the model section describing the calving laws and their background in a bit more detail and in a general context. Furthermore, I think it would be useful to have some discussion towards the end of the paper about how, qualitatively, you expect processes to be affected by e.g. different choice of calving laws, which include different mechanisms, and basal melt (exclusion of it mentioned line 14, section 2). Perhaps insert a separate discussion and conclusion. This would help make the paper more accessible to a general reader who is interested in what the key parameters really are.**

We have included a much more detailed description of the literature on calving in the model section (section 2.2). The text is already given above as one of the responses to referee # 1, but we repeat it here (as before, italics indicate new or altered text)

"...We take the second condition to be a 'calving law'. *While a stress condition is sufficient to close the force balance model (1a), a calving model can be understood as fixing the free boundary location. The next section describes the different choices of calving laws used here.*

**Calving model**

*In our view, the CD model is a cartoon version of the linear elastic fracture mechanics explored in by Weertman (1973,1980) and van der Veen (1998a,b). These papers consider the 'mode 1' (Zehnder, 2012) propagation of vertical cracks into ice under tensile (extensional) stresses. This is done by computing stress levels around the crack tip from known Green's functions for parallel-sided elastic slabs with cracks penetrating from the upper or lower surfaces, accounting*
Interactive
comment

*for the pressure exerted by water in the cracks, and applying a fracture toughness criterion. The CD model by contrast assumes that extensional stress increases with depth in the ice in a linear, cryostatic fashion. The model then computes crevasse penetration as being the distance from the upper and lower surfaces at which that extensional stress becomes sufficiently negative (that is, sufficiently compressive) to overcome the pressure exerted by water at the same depth. The CD model therefore does not compute stress with the same level of sophistication as the papers by Weertman (1973,1980) and van der Veen (1998a,b), but follows the same basic approach of computing crevasse propagation based on a known etWeertman1973,Weertman1980 and **??**, but follows the same basic approach of computing crevasse propagation based on a known ice geometry, extensional stress and crevasse water pressure, and it has the advantage of tractability.*

*The basic method in van der Veen (1998a,1998b) in principle allows for a constraint to be computed that links ice thickness, applied extensional stress, crevasse water level and fracture toughness at the moment that surface and basal crevasses together first penetrate through the entire ice thickness. Given that extensional stress is a function of ice thickness through (1e), this constraint could be converted into a criterion for the thickness $h_c$ at which calving occurs,giving a more sophisticated version of the Nick et al (2010) CD model. However, the papers by van der Veen do not deal with the case in which both, surface and basal crevasses are present and interact with each other (so the relevant Green's functions are not given), and he does not explicitly compute a condition for calving that could be put in the form (1f). As a result, we confine ourselves to the simpler CD model here.*

*One of the practical pitfalls of the CD model is that it predicts no calving at all if $d_w = 0$ and surface crevasses are free of water. It is possible that this is an artifact of the simple representation of stress in the CD model, where the tensile stress driving crevasse propagation is assumed to have the same dependence*

[Figure]

*on depth below the ice surface regardless of whether a crevasse is present or not. In reality, the formation of crevasses that penetrate through a significant fraction of the ice shelf leads to extensional stress becoming more concentrated around the crack tips than for shallow crevasses (see for instance Fig. 4 of van der Veen (1998a)). This represents a positive feedback on crack propagation, and could lead to calving even for the case of water-free surface crevasses (see also Weertman, 1980).*

*More recently, others have extended the linear elastic fracture mechanics approach of Weertman (1973,1980) and van der Veen (1998a,b) to include effects such as the role distributed damage due to the formation of microcracks in initiating crevasse formation, the blunting of cracks tips due to viscous deformation, and the presence of significant torques near the calving front (Krug et al, 2014, Mobasher et al, 2016, Jimenez et al 2016, Yu et al, 2017). The complexity of these processes however makes them difficult to parameterize in a model that does not resolve the scale of individual crevasses, and we do not consider them here.*

*The Nick et al (2010) CD calving model, along with the work of Weertman (1973,1980) and van der Veen (1998a,b), is based on tensile failure. We can contrast this with the shear failure model of Bassis and Walker (2011) (see also Bassis and Jacobs (2013) and Ma et al (2017)).* The CD model requires $d_w > 0$ and predicts calving for any $h$ below the value given by (1f), instantaneously removing all parts of the glacier shelf that are too thin. By contrast, the shear failure model of **?** predicts that calving will start at a critical calving front thickness and not occur below that thickness, so the inequality in (1g) would need to be reversed. ..."

We also make brief reference to this again in the discussion and conclusions section, though it seemed inappropriate to speculate as to the results of using other calving models. Consequently, we have limited ourselves to pointing out that the

CD model for floating calving fronts simply prescribes calving front thickness, independently of bed topography below the shelf, in terms of a calving parameter ($d_w$), and the relationship between flux and depth to bedrock at the grounding line obtained in that case will be the same for any other calving model that also prescribes a calving front thickness independently of depth to bedrock:

"For a floating ice shelf, calving cliff height is simply proportional to crevasse water depth and independent of depth to bedrock. In other words, the CD model can then be thought of as a generic calving model that imposes a fixed thickness at the floating glacier terminus. . . . *As we have indicated, the thickness of floating calving fronts in the CD model is uniquely controlled by the crevasse water depth parameter, and does not depend on depth to bedrock. The same generic relationship between ice flux and depth to bedrock at the grounding line will therefore be obtained for any other calving law that fixes the height of a floating calving front independently of depth to bedrock.* By contrast, the CD model results are unlikely to be robust in the same way for grounded calving fronts. "

As for the effect of melting, we reference melting again in the discussion and conclusions section as a process that will affect results. We have already completed this work and are preparing a second manuscript; unfortunately, incorporating this into the present paper would simply make it unduly long. (The present paper is based on a manuscript previously submitted to a fluids journal. That manuscript covered melting as well as calving, and it was felt that it was not only too specific to glaciology but also too broad in scope, and should be split into papers focusing on calving and on the additional effects of melting.):

"Our aim has not been to be authoritative in establishing the existence of an anomalous flux-depth relationship: our model contains at least *three* components that can be improved upon. . . .

*Second, we have neglected the effect of basal melting on the shelf here. This is tractable in the framework we have developed here with a simple, prescribed*

*basal melt rate, but doing so still introduces sufficient complications to lie beyond the scope of a single paper; a second manuscript that incorporates melting into our theory is in preparation.*"

- **I would also really appreciate a table of variables being included. There were several points in the manuscript where this would have been useful to reference as so many different variables are used.**

  A table is now provided in the supplementary material

- **Abstract, line 3-4 Re-phrase as confusing ordering at the moment. I suggest 'The length of any floating ice shelf present also affects the lateral drag, hence calving is an important process'.**

  Reworded to entire abstrract

*We consider the flow of marine-terminating outlet glaciers that are laterally confined in a channel of prescribed width. In that case, the drag exerted by the channel side walls on a floating ice shelf can reduce extensional stress at the grounding line. If ice flux through the grounding line increases with both, ice thickness and extensional stress, then a longer shelf can reduce ice flux by decreasing extensional stress. Consequently, calving has an effect on flux through the grounding line by regulating the length of the shelf. In the absence of a shelf, it plays a similar role by controlling the height of the calving cliff. Using two calving laws, one due to Nick et al based on a model for crevasse propagation due to hydrofracture, and the other simply asserting that calving occurs where the glacier ice becomes afloat, we pose and analyse a flowline model by two methods: direct numerical solution and matched asymptotic expansions. The latter leads to a boundary layer formulation that predicts flux through the grounding line as a function of depth to bedrock, channel width, basal drag coefficient, and a calving parameter. By contrast with unbuttressed marine ice sheets, we find that flux can decrease with increasing depth to bedrock at the grounding line, reversing the*

*usual stability criterion for steady grounding line location. Stable steady states can then have grounding lines located on retrograde slopes. We show how this anomalous behaviour relates to the strength of lateral versus basal drag on the grounded portion of the glacier, and to the specifics of the calving law used.*

- **Abstract, line 9 'increasing depth to bedrock' - better at this point to refer to as 'retrograde bed slope' as this how people usually think of it?**

  This is not limited to retrograde slopes: if the grounding line moves from shallower to deeper water, regardless of whether the grounding line has to move seaward or inland to achieve that, the flux can decrease rather than increase. We have therefore left the wording as was.

- **Intro, line 19-23 Sentence far too long. Insert full stop after first part. i.e. '...that can alter the flux-to-bedrock-depth relationship. These include...'**

  Changed

- **Intro, line 15 onwards Can you insert a sentence or two justifying choosing these two calving laws over others? Or say you'll do this in model description section and add discussion in there as mentioned above.**

  Reworded as

  "We investigate how two particular calving laws that are relatively widely used in models for tidewater glaciers affect buttressing in a simplified flowline model. *The ice flow model itself* lacks the sophistication of models that resolve the cross-channel dimension. Instead, it relies on a parameterization of lateral drag in terms of the mean along-channel velocity. . . . *The rationale for the calving models used here is described in greater detail in section 2.2.* . . . "

- **page 3, line 10 Why using B, rather than more standard choice of A for Glen coefficient?**

From the continuum mechanics perspective, writing stress as a function of strain rate is more natural than the other way around (because that's how a Stokes flow model is usually written, as a second order elliptic problem for velocity). Given that, using $A$ introduces unneccessary powers of $n$.

- **page 3, line 24 Can you extend discussion here with a couple of sentences about limitations of this parameterisation?**

The description of this parameterization and its limitations go all the way to equation (1e) on page four, and we refer to the discussion in Pegler (2016) for more detail. We are also attempting to be honest in describing the heuristic nature of flowline models of this kind, both here, in the introduction and the discussion and conclusions. To reiterate the point made in the response to referee # 1, the point is that this kind of flowline model has been widely used elsewhere — so the ability to interpret existing results remains important — that it is based on the basic physics involved, and furthermore, that the model allows rapid computation over large sets of parameter values). The alternative would be to use a much more sophisticated and costly model; proceeding at "face value" is done in the hope that something *useful* can be learnt without resorting to that alternative, which would make it much more difficult to explore parameter space to the extent we are able to, and to come to *qualitative* conclusions. Of course, one could assert that only the best type of model should ever be used and that flowline models are dead. We would respectfully disagree, and point to simple box models in glaciology and elsewhere as a tool still used to develop understanding. Our effort should be seen as being in that spirit, but situated somewhere between box models and sophisticated three-dimensional multiphysics models.

To expand on where we address the rationale for the model (in addition to the text in section 2.1, from "The second term $\bar{B}' w^{-1/n-1} h|u|^{1/n-1u}\ldots$" to equation (1e)), we have in the introduction

"We investigate how two particular calving laws that are relatively widely used in

models for tidewater glaciers affect but- tressing in a simplified flowline model. The model lacks the sophistication of models that resolve the cross-channel dimension. Instead, it relies on a parameterization of lateral drag in terms of the mean along-channel velocity (Dupont and Alley, 2005; Nick et al., 2010; Jamieson et al., 2012; Hindmarsh, 2012; Pegler et al., 2013; Robel et al., 2014, 2016; Pegler, 2016). The chief advantages of the model are that it allows flux through the grounding line to be computed rapidly as a function of ice thick- ness through the use of a boundary layer theory (Schoof, 2007a) and that the role of different physical mechanisms becomes comparatively easy to trace. Future work will be required to address whether our results are reproduced qualitatively by more sophisticated (and more computationally intensive) models, and we hope that this paper can motivate such work."

and in the conclusions

"Our aim has not been to be authoritative in establishing the existence of an anomalous flux-depth relationship: our model contains at least two components that can be improved upon. First, the parameterised description of lateral drag should eventually be dispensed with, replacing our model with one that resolves the cross-channel dimension. The scaling that underlies our boundary layer model should still be applicable in that case, but the actual boundary layer model will consist of a set of coupled partial differential equations (as opposed to ordinary differential equations) and is likely to be much more onerous to solve for a large number of parameter combinations, as we have been able to do here."

Note that this is not new text, but hopefully addresses the point adequately; we feel that anything else we might say would amount to repetition of the material already in the paper.

- **Figure 1 Table of variables would certainly help reader when looking at this figure. Also, for a grounded terminus shouldn't inequality for h actually read hf $\leq$ hc as still grounded if calving happens at flotation. Alternatively,**

**perhaps you could insert a third lower diagram illustrating hf = hc = hg since this could then be used as a reference when describing the second calving law you use (page 5, lines 1-2)?**

A table has been provided at the end of section 2. Inequality has been changed — though we left the figure as was with two lower panels in order not to crowd things.

- **Figure 2 Grey shaded regions do not show at all when printed. Make darker. Figure 2 Insert space 'Panel (b)' in caption**

  Done

- **equation 1h I think this is the first time dw is used but you do not explicitly state that it is water depth. At least have in table.**

  Corrected and included in table

- **equations 1h/1i Can you line these up properly so 'at and if' are in line (and do similarly at several other points in paper).**

  We are using the standard

  ```
  \begin{align} ... \end{align}
  ```

  environment, which the TCD documentclass seems to align in the fashion shown here. That appears to be a Copernicus / cryosphere style and documentclass issue, and beyond our control.

- **page 7, lines 1-12 As mentioned above I think you want more discussion and context here. Given the length of this section I think it would also be helpful to split section 2 up into a couple of subsections e.g. 'ice flow model', 'calving laws'. This would help remind reader where to reference back to when thinking about the different parameterisations later.**

We have split this into subsections, see above for discussion and context.

- **section 3.1 a few more words reminding us what equations are e.g. page 8 line 5 'if $rh \geq b$ i.e. flotation.**

  To avoid cumbersome extra text inline with the equation, we have now prefixed equation (4c) by

  "*with $\theta$ the indicator function for flotation*"

- **page 9, line 9 '...glacier terminus, which are of the form'**

  Changed

- **page 9, line 14 change to 'monotonically downward'**

  'downward monotonically' is not ok? Especially if it's 'monotonic in $x$'?

- **page 11, line 24ish confusing now having 'B' as a rescaled b and having the B etc earlier for Glen. Preferably change Glen to A but at least introduce table**

  Included in table — this was one of the reasons for using $\bar{B}$ rather than just the straightforward $B$

- **page 12, line 27 Physical interpretation of $\lambda$ being small?**

  We've restructured this and moved the rescaling in $\lambda$ up in the text, to say

  "*In order for the rescaling in $H$ above to be consistent, we also require that the calving front thickness be similarly small. This turns out to require that $\lambda \sim O(\varepsilon^{n^2/(n+1)^2})$, and we define*

  $$\Lambda = r^{-1}\varepsilon^{-n^2/(n+1)^2}\lambda$$

  *assuming that $\Lambda = O(1)$; all this implies is that water depths in surface crevasses are not so large as to create calving cliff heights much larger than the expected depth to bedrock at the grounding line.* "

- **page 13, line 4-8 Long sentence, difficult to take in. Split up.**

  Split at 'first', 'second' and 'third':

  "*Physically, the first condition states that the flux near the grounding line in the 'outer' problem is the flux that enters the boundary layer at its upstream end. The second condition states that near that upstream end of the boundary layer, extensional stress gradients have become insignificant and flux is given by a shallow ice type formula (with $U = Q/H$, the condition can be re-written as $Q \sim -H|H_X|^{n-1}H_X$, the appropriate local-force-balance formula in our case). Lastly, the third condition states that velocities in the interior of the boundary layer are large compared with those in the rest of the glacier.*"

- **page 14, line 1-2 'on the then-redundant parameter $\Lambda$ to write alternatively' -> 'on the now-redundant parameter $\Lambda$ to simplify the expression to'**

  Changed as suggested

- **page 14, line 25 'the CD model produces the same result as..., which is reassur- ing'**

  Not sure whether 'reassuring' is the right word, that seems subjective

- **page 17, eqn 11 insert fullstop.**

  Done

- **page 22, line 13ish Sentence between the two equations (line numbering gone askew here) should read 'Integrating and applying the boundary condition shows that extensional stress...'**

  Corrected

- **page 22, eqn 23 insert fullstop.**

  Done

- **page 25, line 16 You seem to sometimes talk in terms of the rescaled $B$ (like here) and at other points in terms of depth to bedrock $b$ (e.g. line 20). I would stick to the variable $b$?**

We have tried to correct all instances of $B$ in the discussion

- **page 26, line 16 onwards Yes, this and the following discussion are good points but then the paper ends rather suddenly. Could you put this into a bit more context and suggest extensions/alternative approaches.**

We are not entirely sure how to address this point. What is 'this' in particular — the dependence of calving cliff height on water depth as a control parameter? We have elaborated a little on this, as follows:

"We have chosen to take the calving model at face value, simply prescribing the crevasse water depth control parameter. This is worth emphasizing as the dependence of calving cliff height on flotation thickness predicted by the calving model turns out to be key to the anomalous flux-depth relationship. *It is likely that other, more sophisticated calving models (for instance one based on the formulation in van der Veen (1998a,1998b) can also be written in the form of a calving cliff height as a function of crevasse water depth, though presumably with a different specific from the CD model: as in the latter, surface hydrology becomes a key component in understanding calving.*"

- **page 27, line 8 'may be possible at least in principle' - awkward wording.**

Probably not just awkward but downright incorrect — an 'and' was missing

"*Since an anomalous flux-depth-to-bedrock relationship may be possible **and** would have significant consequences for stable outlet glacier configurations, and it may be worth testing this before embarking on simulations of actual glaciers using different calving laws.*"

- **page 27, line 12 'At issue' change to 'An important issue'**

We believe 'at issue' to be correct: Collins dictionary defines the phrase as meaning 'The question or point at issue is the question or point that is being argued about or discussed.' — in this case, hopefully, discussed.

- **page 28, line 13 'meaning a functional relationship' change to 'giving a functional relationship'**

Corrected

**3    Reviewer # 3**

- **If the Nick et al crevasse depth calving model is to be referred to as 'CD' (as in Nick et al 2010 IIRC) then why not have a similar acronym for the other model (I think Nick called it 'FL'?)**

  We have relabelled 'calving at flotation' as the 'FL' calving law throughout the text.

- **P5-6, eqns 1h,1i: Although the derivations of these expression is in the supplement, would it not make sense to say at this point that the arise from combining the relation- ship between stress and thickness at the calving front and the relationship between 'dry' crevasse depth and stress.**

  As part of other revisions, we have reworded this as

  "In our view, the CD model is effectively a simpler version of the linear elastic fracture mechanics explored in by Weertman (1973,1980) and van der Veen (1998a,b). These papers consider the 'mode 1' (Zehnder, 2012) propagation of vertical cracks into ice under tensile (extensional) stresses. This is done by computing stress levels around the crack tip from known Green's functions for parallel-sided elastic slabs with cracks penetrating from the upper or lower surfaces, accounting for the pressure exerted by water in the cracks, and applying a fracture toughness criterion to determine whether a crack will propagate. The CD model by contrast assumes that extensional stress increases with depth in the ice in a linear, cryostatic fashion. The model then computes crevasse penetration as being the distance from the upper and lower surfaces at which that extensional stress becomes sufficiently negative (that is, sufficiently compressive) to overcome the pressure exerted by water at the same depth. The CD model therefore does not compute stress with the same level of sophistication as the papers by Weertman (1973,1980) and van der Veen (1998a,b), but follows the same basic approach of computing crevasse propagation based on a known

ice geometry and known water pressures applied inside the crevasses, and it has the advantage of tractability."

- *P6, L15: I don't dispute that sensitivity to dw (and the requirement that dw âĹij h) is a problem for CD: in fact I would go further and say it that in current applications it might be standing in for physics that has nothing to do with hydrology at all.*

We are inclined to agree. The main question is whether the dependence of $h_c$ on $b$ is appropriate, given that $d_w$ may be a degree of freedom that is difficult to pin down. We refer to this briefly in the discussion and conclusions section, where we point out that (new text in bold face)

"For a floating ice shelf, calving cliff height in the CD model is simply proportional to crevasse water depth and independent of depth to bedrock. In other words, the CD model can then be thought of as a generic calving model that imposes a fixed thickness at the floating glacier terminus. . . .

*As we have indicated, the thickness of floating calving fronts in the CD model is uniquely controlled by the crevasse water depth parameter, and does not depend on depth to bedrock. The same generic relationship between ice flux and depth to bedrock at the grounding line will therefore be obtained for any other calving law that fixes the height of a floating calving front independently of depth to bedrock. By contrast, the CD model results are unlikely to be robust in the same way for grounded calving fronts.*"

- **P9, L24: 'Our aim in what follows. . . ' rather than a single sentence, it might be helpful to quickly sketch out the line of thought. It was not until about P15 that I got the sense of that.**

We have expanded this as follows:

"Our aim in what follows is to explain the results in *Figs. 3–4* using the same boundary layer approach as in Schoof (2007b). *In particular, we will show that*

*flux through the grounding line can be computed to leading order in the parameter $\varepsilon$ as a function depth to bedrock and channel width at the grounding line, as well as of the calving parameter $\lambda$, friction coefficient $\gamma$ and the remaining physical parameters ($r$, $m$, $n$). Given such a relationship, it is then possible to determine how the grounding line location in a steady state depends on accumulation rates, purely by balancing net accumulation over the domain with outflow of ice through the grounding line.*"

- **P12,L15: 'Despite working at leading order in we have retained two terms that contain factors of in (9) '. Slightly odd phrasing, which might give the impression that the terms are retained even though they are $\ll O(1)$ ? Both factors are (for the case $n = 3$, $m = 1/3$ $\varepsilon$-9/16 $\gg O(1)$. In the paragraph that requires $\gamma$ and $\lambda \ll O(1)$ so for all terms to appear at the same order.**

Good point. We have restructured and reworded this as follows: First, the rescaling in $\lambda$ has been moved up in the text, to say

"*In order for the rescaling in $H$ above to be consistent, we also require that the calving front thickness be similarly small. This turns out to require that $\lambda \sim O(\varepsilon^{n^2/(n+1)^2})$, and we define*

$$\Lambda = r^{-1}\varepsilon^{-n^2/(n+1)^2}\lambda$$

*assuming that $\Lambda = O(1)$; all this implies is that water depths in surface crevasses are not so large as to create calving cliff heights much larger than the expected water depth at the grounding line.*"

and thereafter we deal with the rescaling in $\gamma$ by saying

"*In order to make the balances in (9) work with $\varepsilon \ll 1$, we have to deal with the remaining coefficient that contains a power of $\varepsilon$ in (9). We now make further assumptions about the physics of the flow near the grounding line.*"

The fact that the power of $\varepsilon$ is negative is made clear later, where we state

" We confine our analysis to parameter regimes where this is the case. Note that with $m > 0$ and $n \geq 1$, this implies strictly speaking that $\gamma \ll 1$, and basal friction upstream of the boundary layer is formally small in the parameter regime we are considering. "

- **P13, eq 9g. Does the factor** $|h_x|^{1/m}$-1 **arise in general? Or just because m = 1/n? Perhaps I missed a trick here, but if** $\gamma$ **and** $b_x$ **are small in (7a), the flux expression just depends on the wall drag and driving stress (so** $m$ **does not enter).**

No, that is an apparently careless error, the expression should be

$$Q \sim \lim_{x \to x_g^-} (-w^{n+1} h |h_x|^{n-1} h_x)$$

in agreement with the

$$Q \sim -W^{n+1} H |H_X|^{n-1} H_X$$

in the next paragraph (where a $W^{n+1}$ was also missing). The error does not propagate further, however.

- **P19: typo?** $H'_{f0} \mapsto H'_f$ **in expressions above (14)**

Indeed. Corrected.